# Morphology, Taxonomy, Culm Internode and Leaf Anatomy, and Palynology of the Giant Reed (*Arundo donax* L.), Poaceae, Growing in Thailand

**DOI:** 10.3390/plants12091850

**Published:** 2023-04-30

**Authors:** Chatchai Ngernsaengsaruay, Buapan Puangsin, Nisa Leksungnoen, Somwang Khantayanuwong, Pichet Chanton, Thirawat Thaepthup, Paweena Wessapak, Rumrada Meeboonya, Piyawan Yimlamai, Kapphapaphim Wanitpinyo, Korawit Chitbanyong, Tushar Andriyas, Nattapon Banjatammanon

**Affiliations:** 1Department of Botany, Faculty of Science, Kasetsart University, Chatuchak, Bangkok 10900, Thailand; 2Biodiversity Center Kasetsart University (BDCKU), Bangkok 10900, Thailand; 3Department of Forest Products, Faculty of Forestry, Kasetsart University, Chatuchak, Bangkok 10900, Thailand; 4Department of Forest Biology, Faculty of Forestry, Kasetsart University, Chatuchak, Bangkok 10900, Thailand; 5College of Allied Health Sciences, Suan Sunandha Rajabhat University, Mueang, Samut Songkhram, Bangkaeo 75000, Thailand; 6Department of Music, Faculty of Humanities, Kasetsart University, Chatuchak, Bangkok 10900, Thailand

**Keywords:** Arundineae, *Arundo donax*, clarinet reed, C3 grass, Gramineae, grass anatomy, grass taxonomy, lectotype, pollen morphology, riverbanks

## Abstract

In this paper, we present the morphology, taxonomy, anatomy, and palynology of *Arundo donax*. A detailed morphological description and illustrations of the species are provided, along with information about the identification, distribution, the specimens examined, habitat and ecology, the International Union for Conservation of Nature (IUCN) conservation assessment, phenology, etymology, vernacular name, and uses. The species can be distinguished by its large, tall rhizomatous perennial reed; cauline leaves; an open, large, plumose panicle inflorescence; subequal glumes as long as the spikelets; glabrous rachilla; all bisexual florets; and a lemma with a straight awn and with long white hairs outside below the middle part. In this study, two names were lectotypified: *Arundo bifaria* and *A. bengalensis*, which are synonyms of *A. donax*. The culm internodes in the transverse section have numerous vascular bundles scattered in the ground tissue, and the parenchyma cells have significantly lignified cell walls. Vascular bundles are composed of phloem and xylem and are enclosed in a continuous sclerenchymatous bundle sheath. The chloroplasts in the transverse section of the leaf blades are found only in the mesophyll cells but are absent in the bundle sheath cells, which indicates that it is a C3 grass. The leaves have stomata on both surfaces and are confined to the intercostal zones. The stomata are typically paracytic, with two lateral subsidiary cells placed parallel to the guard cells. The stomatal density is higher on the abaxial surface [450–839/mm^2^ (606.83 ± 72.71)] relative to the adaxial surface [286–587/mm^2^ (441.27 ± 50.72)]. The pollen grains are spheroidal or subspheroidal [polar axis length/equatorial axis length ratio (P/E ratio) = 0.89–1.16 (1.02 ± 0.07)] with a single pore surrounded by a faint annulus, and the exine sculpturing is granular.

## 1. Introduction

*Arundo* L. is a group of large, tall rhizomatous perennial reeds and the small genus in the tribe Arundineae Dumort. under the subfamily Arundinoideae Kunth ex Beilschm. (see detailed descriptions of GPWG [1] and Hardion et al. [2]) and the family Poaceae Barnhart (Gramineae Juss., nom. alt.) [3,4]. Arundinoideae is one of the small grass subfamilies, with 14 genera and 36 species in three tribes with some biogeographical patterns: (1) tribe Arundineae, the most heterogeneous tribe with four genera and 17 species, including Eurasian *Arundo*, Australian *Amphipogon* R. Br. and *Monachather* Steud., and South African *Dregeochloa* Conert; (2) tribe Molinieae Jirásek with four genera and seven species, including *Hakonechloa* Makino ex Honda (one species in Japan), *Molinia* Schrank (one species in Eurasia and Africa), *Moliniopsis* Hayata (one species in Southeast China, Korea, and Japan), and *Phragmites* Adans. (four species, cosmopolitan); and (3) tribe Crinipedeae Hardion with six genera and 12 species, including *Crinipes* Hochst. (four species in East Africa), *Elytrophorus* P. Beauv. (two species in Africa, Asia, and Australia), *Leptagrostis* C. E. Hubb. (one species in East Africa), *Piptophyllum* C. E. Hubb. (one species in South Africa), *Pratochloa* Hardion (one species in South Africa), and *Styppeiochloa* De Winter (three species in South Africa and Madagascar) [2,3,4,5]. Arundineae contains 17 species in four genera: *Amphipogon* (nine species in Australia), *Arundo* (five species in tropical Eurasia), *Dregeochloa* (two species in South Africa), and *Monachather* (one species in Australia) [2,4]. *Arundo* consists of five accepted species and is distributed from the Mediterranean to temperate East Asia and the Philippines: three species in the Mediterranean, i.e., *Arundo donaciformis* (Loisel.) Hardion, Verlaque, and B. Vila (Southern France to Northwest Italy), *A. micrantha* Lam. (widely distributed), and *A. plinii* Turra (Italy to West Crete), one species in Nansei-shoto (Ryukyu Islands) to Taiwan and the Philippines, i.e., *A. formosana* Hack., and one species in West and Central Asia to temperate East Asia, i.e., *A. donax* L. It grows primarily in the subtropical biome [5].

The culms of *Arundo donax* are a useful source of cane for light construction and for making woodwind instrument reeds [6,7]. Clarinet and other woodwind instruments use a thin strip of cane (culm internode), called a reed, to produce their sound. The reed is placed within the instrument’s mouthpiece, where it vibrates according to the blowing pressure generated by the player. The anatomical characteristics of the culm internodes affect the musical performance of clarinet reeds made from *A. donax* [8]. Reeds for clarinet, oboe, and bassoon are mainly produced from the giant reed, *A. donax*. The best reed canes grow only in a few areas of the Var in France due to its very mild Mediterranean climate with lots of sunshine and the mistral [9].

There have been many previous studies of Poaceae in Thailand. However, identifications often rely on the literature [6,7,10,11,12,13,14,15,16,17,18,19,20], and this is the case for *Arundo*, which has never been revised before for Thailand. In this paper, we provide knowledge on the *Arundo* in Thailand obtained from the research subproject entitled “Morphology, Taxonomy, Anatomy, Palynology, Distribution, and Ecology of *Arundo. donax* L. (Poaceae) in Thailand” under the research project entitled “Study and Quality Development of Reed in Thailand for Clarinet Reed Production and their Utilization in Music Industry”.

## 2. Results

### 2.1. Morphology and Taxonomy

#### 2.1.1. *Arundo* L., Sp. Pl. 1: 81. 1753

*Arundo* L., Sp. Pl. 1: 81. 1753.—*Donax* [non Loureiro 1790: 11 (Marantaceae)] P. Beauv., Ess. Agrostogr.: 77. 1812, nom. illeg., nom. superfl.—*Donacium* Fr., Bot. Not. 132. 1843, nom. superfl.—*Eudonax* Fr., Bot. Not. 132. 1843, nom. superfl. Type species: *Arundo donax* L.

*Amphidonax* Nees in Lindley, Nat. Syst. Bot., ed. 2: 449. 1836. Type: *Amphidonax bengalensis* (Retz.) Nees ex Steud. = *Arundo donax* L.

*Scolochloa* [non-Link 1827: 136, nom. cons.] Mert. and W. D. J. Koch, Deutschl. Fl., ed. 3, 1: 374, 528. 1823, nom. rej., nom. superfl.

Arundineae are characterised by glumes that are typically as long as or longer than the lowest floret, while Molinieae and Crinipedeae generally have glumes that are shorter than the lowest floret [3,4]. Only one genus of each tribe is found in Thailand, i.e., *Arundo*, *Phragmites*, and *Elytrophorus*, respectively.

*Arundo* can be distinguished by its large, tall rhizomatous perennial reed; cauline leaves; open, large, plumose (feather-like), panicle inflorescences; subequal glumes, as long as spikelets; glabrous rachilla; all bisexual florets; lemma with a straight awn and with long white hairs outside below the middle part. A comparison of the morphological characteristics of these three related genera is described from herbarium specimens and from the author’s observations during the fieldwork and is also taken from the literature [6,7,10,11,12,13,14,15,16,17,18,19,20] (Table 1).

#### 2.1.2. *Arundo donax* L., Sp. Pl. 1: 81. 1753 (Figure 1, Figure 2, Figure 3, Figure 4 and Figure 5)

*Arundo donax* L., Sp. Pl. 1: 81. 1753; Hook. f., Fl. Brit. India 7: 302. 1897; H. L. Blomq., Grasses N. Carolina: 56, fig. 56. 1948; Bor, Grasses Burma, Ceylon, India, and Pakistan 1: 413, fig. 44. 1960; Y. N. Lee, Man. Korean Grasses: 236, fig. 41. 1966; Backer and Bakh. f., Fl. Java (Spermatoph.) 3: 526. 1968; Gould, Grass Syst.: 308, fig. 5-116. 1968; Gilliland et al. in H. M. Burkill, Fl. Malaya 3: 51. 1971; C. C. Hsu in H. L. Li et al., Fl. Taiwan 5: 382, t. 1368. 1978; Lazarides, Phanerog. Monogr. 12: 151. 1980; R. W. Pohl in W. C. Burger, Fl. Costaric. 4: 65, fig. 17. 1980; Cope in Nasir and S. I. Ali, Fl. Pakistan 143: 21, fig. 2. 1982; T. Koyama, Grasses Japan: 237, fig. 87. 1987; Duist., Suppl. Gard. Bull. Singapore 57: 34, fig. 21. 2005; S. L. Chen et al. in C. Y. Wu, P. H. Raven and D. Y. Hong, Fl. China 22: 447. 2006.—*Cynodon donax* (L.) Raspail, *Ann. Sci. Nat*. (Paris) 5: 302. 1825.—*Scolochloa donax* (L.) Gaudin, Fl. Helv. 1: 202. 1828.—*Donax donax* (L.) Asch. and Graebn., Fl. Nordostdeut. Flachl. 101. 1898, not validly publ.—*Arundo latifolia* Salisb., Prodr. Stirp. Chap. Allerton 24. 1796, nom. illeg., nom. superfl.—*Donax arundinaceus* P. Beauv., Ess. Agrostogr. 152 (as *D. arundinacea*), 161, t. 16, fig. 4. 1812.—*Scolochloa arundinacea* (P. Beauv.) Mert. and W. D. J. Koch, Deutschl. Fl., ed. 3, 1: 530. 1823, nom. illeg., nom. superfl. Type: Spain, *Anon s.n.* (lectotype L [Herb. A. van Royen 912.356-93], designated by S. A. Renvoize in Jarvis et al., Regnum Veg. 127: 21. 1993.

*Arundo sativa* Lam., Fl. Franç. 3: 616. 1778.—*Donax sativus* (Lam.) C. Presl, Cyper. Gram. Sicul. 32. 1820. Type: not known.

*Arundo bifaria* Retz., Observ. Bot. 4: 21. 1786.—*Donax bifarius* (Retz.) Trin. in *Spreng., Neue Entdeck*. Pflanzenk. 2: 73. 1821.—*Amphidonax bifaria* (Retz.) Nees ex Steud., Syn. Pl. Glumac. 1: 197. 1854. Type: India, s.d., *Wight 1748* (*Wight collection, East India Company Herbarium 5018A*) (lectotype K-W [K001104515!], isolectotypes BM [BM000949265!], E [E00576404!, E00576405!, E00576406!, E00576407!], K [K000032473!], designated here) (Figure 6).

*Arundo bengalensis* Retz., Observ. Bot. 5: 20. 1788.—*Aira bengalensis* (Retz.) J. F. Gmel., *Syst. Nat*., ed. 13. 2(1): 174. 1791.—*Donax bengalensis* (Retz.) P. Beauv., Ess. Agrostogr.: 78. 1812.—*Amphidonax bengalensis* (Retz.) Nees ex Steud., Syn. Pl. Glumac. 1: 197. 1854. Type: India, Bengal, 1820, *Wallich Cat. 5018D* (*East India Company Herbarium 5018D*) (lectotype K-W [K001104517!], designated here) (Figure 7).

*Arundo longifolia* Salisb. ex Hook. f., Fl. Brit. India 7: 303. 1897, pro syn. Type: not known.

*Arundo triflora* Roxb. ex Hook. f., Fl. Brit. India 7: 303. 1897, pro syn. Type: not known.

*Arundo bambusifolia* Hook. f., Fl. Brit. India 7: 303. 1897. Type: not known.

*Arundo glauca* Bubani, Fl. Pyren. 4: 303. 1902, nom. illeg.

*Lectotypifications*. *Arundo bifaria* was named by Retzius based on the specimen from India (originally “Habitat ad margines ftagnorum et foffarum in India Orientali” in the first publication) but did not mention the collector number or the herbaria in which they were present [21]. We located the specimen *Wight 1748* [*Herb. Wight, East India Company Herbarium (EICH) 5018A*], which matches the type of *A. bifaria* collected from India. We found seven sheets of this specimen: one sheet at K [K000032473], one sheet at K-W [K001104515], one sheet at BM [BM000949265], and four sheets at E [E00576404, E00576405, E00576406, E00576407], and following Art. 9.6 of the ICN [22], they constitute syntypes. Therefore, the complete and well-preserved specimen *Wight 1748* at K-W [K001104515] is selected here as the lectotype, following Art. 9.3 and 9.12 of the ICN [22] (Figure 6).

*Arundo bengalensis* was named by Retzius based on the specimen from Bengal, India (originally “Habitat in Bengala” on the first publication) but did not mention the collector number or the herbaria in which it was present [23]. We traced one sheet of the specimen *Wallich Cat. 5018D* (*EICH 5018D*) at K-W [K001104517], collected from Bengal, India (originally “Bengala inferior 1820” on the label), and following Art. 9.6 of the ICN [22], it constitutes a syntype. It is selected here as the lectotype, following Art. 9.3 and 9.12 of the ICN [22] (Figure 6). 

*Description*. *Habit*: large, tall perennial rhizomatous grass, loosely tufted, 3–8.2 m tall (including inflorescence). *Rhizomes* are horizontal, variously shaped, 3.5–5 cm thick, fleshy when young, woody when mature; scale leaves are scarious (scariose), brown. *Adventitious roots* are 1–3 mm in diameter. *Shoots* arising from the rhizome. *Culms* erect, cylindrical (terete), gradually narrowing towards the apex, with numerous nodes and internodes, green, turning greenish yellow when mature, stramineous when dry, herbaceous (soft, non-woody tissue) when young, turning woody when mature, glabrous, glossy; culm nodes solid; culm internodes (35–)55–125 or more, hollow, 1.7–17 cm long, 1–5 lowermost internodes usually shorter than the other; basal culm internodes 2.5–17 cm long, 1.5–3.3 cm in diameter; middle culm internodes 4.7–15.7 cm long, 1.3–2.9 cm in diameter; apical culm internodes 1.7–10.5 cm long, 0.4–2.5 cm in diameter; basal culm internode walls 1.4–7.4 mm thick; middle culm internode walls 0.9–3.8 mm thick; apical culm internode walls 0.3–1.9 mm thick; intravaginal branching. *Leaves* simple, cauline, distichous (two ranked); leaf blades flat, green, base pale yellow, spreading, apical part recurved (curved downward), linear or linear-lanceolate, 24–73.2 × 2–11.7 cm, gradually narrowing towards the apex, apex long acuminate, margins scabrous, both surfaces glabrous and glaucous, midrib grooved above, raised below, veins numerous, parallel, distinctly on both surfaces; auricles pale yellow, outside hairy; ligules membranous, 1–2.3 mm long, with minutely ciliolate margin; collars pale yellow, hairy; leaf sheaths pale green, overlapping, tightly clasping the culm, persistent, cylindrical, 7–13.5 cm long, both surfaces glabrous; branch leaves are usually smaller than cauline leaves, 12–55 × 1–7 cm; young leaf blades are involute (inrolled). *Inflorescence* an open large, plumose (feather-like) panicle, terminal, purplish, turning pale brown with age, 70–165 cm long (including peduncle), 42–80 × 17–54 cm (excluding peduncle); rachis (central axis) angular, with sharp or obtuse angles, glabrous; primary branches 32–48, spiral or pseudowhorl, (5–)10–43 cm long, erect or ascending, apical part usually recurved; peduncle cylindrical, 26–95 cm long, basal part 0.7–2.1 cm in diameter, middle part 0.7–1.8 cm in diameter, glabrous. *Spikelets* solitary, laterally compressed, lanceolate or narrowly elliptic, 6–16 × 1.5–6 mm; pedicel slightly flattened, 1.5–10 mm long, scabrous; glumes two, persistent, lower glume and upper glume similar, lanceolate or linear-lanceolate, 5–13.5 × 0.6–1.6 mm, subequal, apex acute or acuminate, margins entire, membranous or chartaceous, glabrous, three or five veined; rachilla glabrous, disarticulating above the glumes and between the florets. *Florets* 2–3(–4), all bisexual, similar, falling off at maturity, the lowest floret larger than the other (gradually decreasing in size); lemma lanceolate or linear-lanceolate, 4–14.5 (including awn) × 1–1.8 mm, apex two toothed, with a straight awn arising from sinus, hyaline, with dense long white hairs outside and below the middle part, up to 6 mm long, three or five veined, midvein protrude in 1–5 mm long awn; palea narrowly elliptic or lanceolate, 2.5–6 × 0.5–1.3 mm, apex acute, margins folded, with a keel on each side, ciliolate on keel, hyaline, two veined. *Flowers*: lodicules 2, 0.1–0.2 mm long; stamens three, filaments 1–2 mm long, anthers yellow, 1.3–2 × 0.3–0.7 mm; ovary superior, ellipsoid, or narrowly ellipsoid, 0.3–0.7 × 0.1–0.2 mm, glabrous; style two, free, 0.5–1.2 mm long; stigma plumose, 1–1.7 × 0.2–0.5 mm. *Fruits* not seen. Measurements of the vegetative and reproductive parts of *Arundo donax* are presented in Table 2.

*Recognition*. *Arundo donax* is characterised as a large, tall rhizomatous perennial reed, loosely tufted, 3–8.2 m tall (including inflorescence), bamboo-like; culms with numerous nodes and internodes, woody when mature; hollow culm internodes; cauline leaves, distichous (in two vertical ranks or rows on opposite sides of the culm); linear or linear-lanceolate leaf blades, 24–73.2 × 2–11.7 cm; open large, plumose, panicle inflorescences, purplish, turning pale brown with age; spikelets solitary, laterally compressed, with 2–3(–4) florets, all bisexual; similar, subequal persistent glumes; lemma two toothed at apex, with a straight awn arising from sinus and with dense long white hairs (silky hairs) outside below the middle part.

**Figure 1 plants-12-01850-f001:**
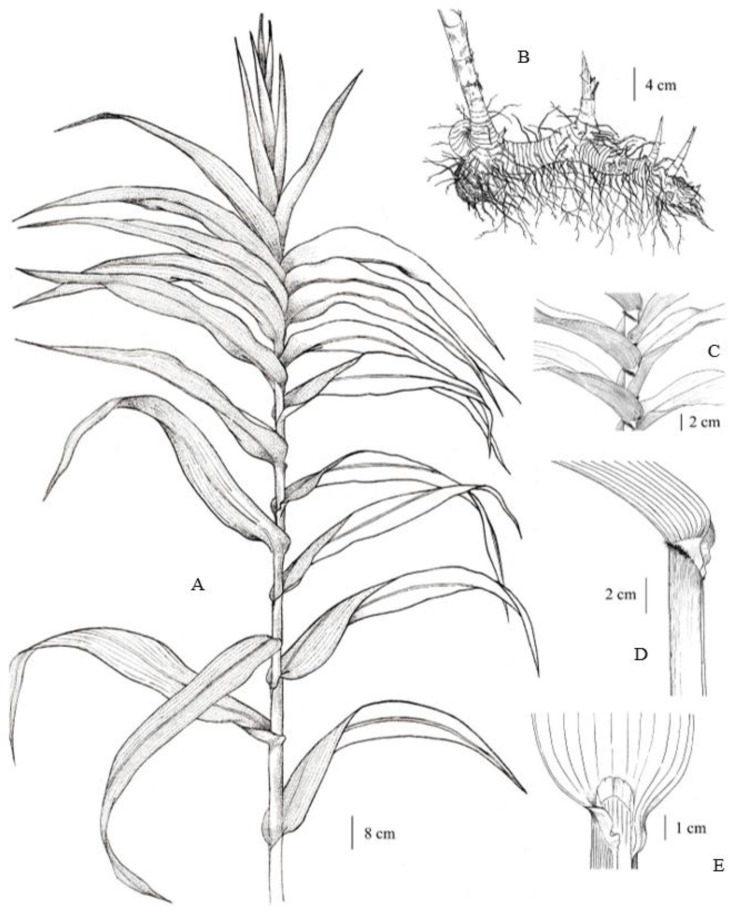
*Arundo donax*. (**A**) distichous cauline leaves with leaf sheaths overlapping, tightly clasping the culm; (**B**) rhizome, young shoots, and adventitious roots; (**C**) collars with hairs; (**D**) leaf sheath, collar, and auricles; and (**E**) membranous ligule. Drawn by Paweena Wessapak.

**Figure 2 plants-12-01850-f002:**
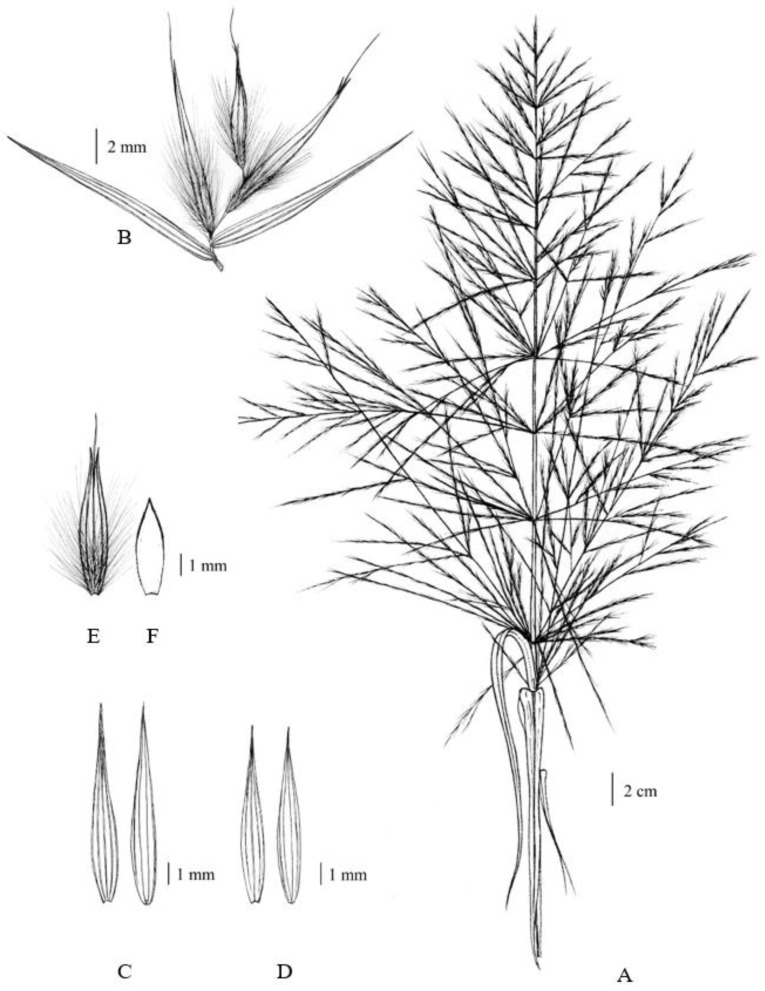
*Arundo donax*. (**A**) an open large, plumose, panicle inflorescence; (**B**) spikelet with glumes and florets; (**C**) lower glumes (outside and inside); (**D**) upper glumes (outside and inside); (**E**) lemma two-toothed at the apex, with a straight awn arising from the sinus, and with dense long white hairs outside below the middle part; and (**F**) palea. Drawn by Paweena Wessapak.

**Figure 3 plants-12-01850-f003:**
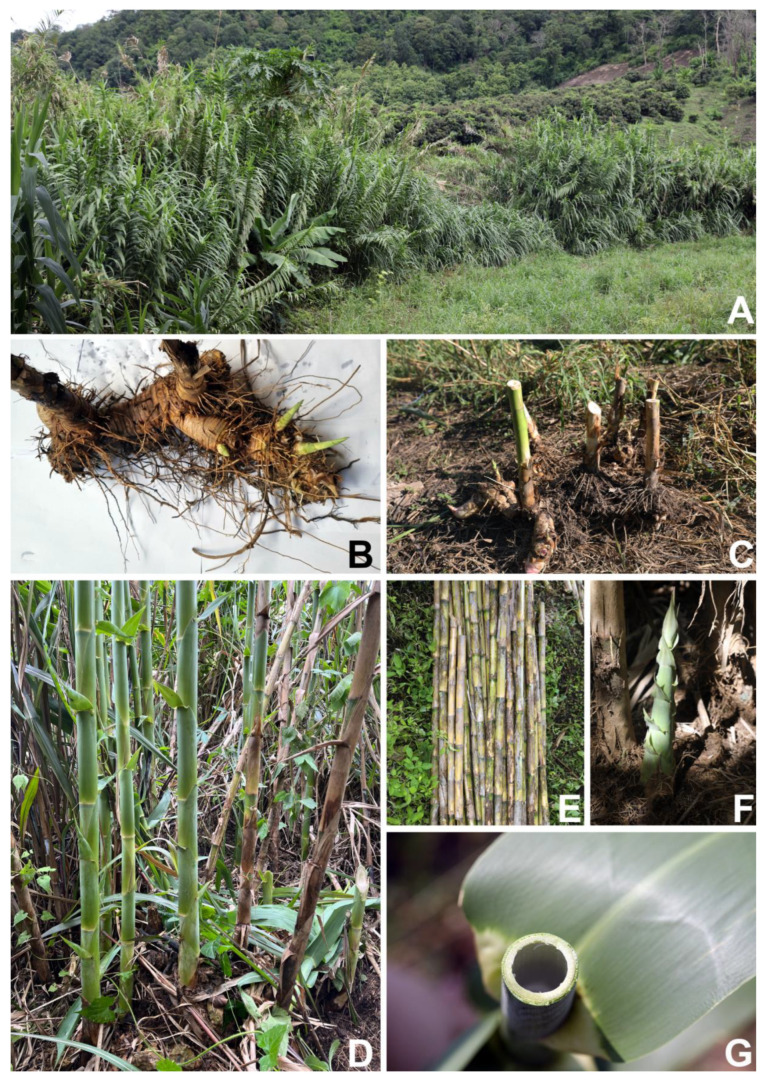
*Arundo donax*. (**A**) Habitat and habit (in an open area along Huai Mae O, Ban Sop O, Mae Na Subdistrict, Chiang Dao District, and Chiang Mai Province); (**B**,**C**) rhizomes, young shoots, and adventitious roots; (**D**) young and mature culms showing nodes, internodes, and leaf sheaths; (**E**) mature culms showing nodes and internodes (persistent leaf sheaths removed); (**F**) young shoot; and (**G**) culm internode with a hollow central pith. Photos: (**A**,**C**–**G**) Chatchai Ngernsaengsaruay; (**B**) Korawit Chitbanyong.

**Figure 4 plants-12-01850-f004:**
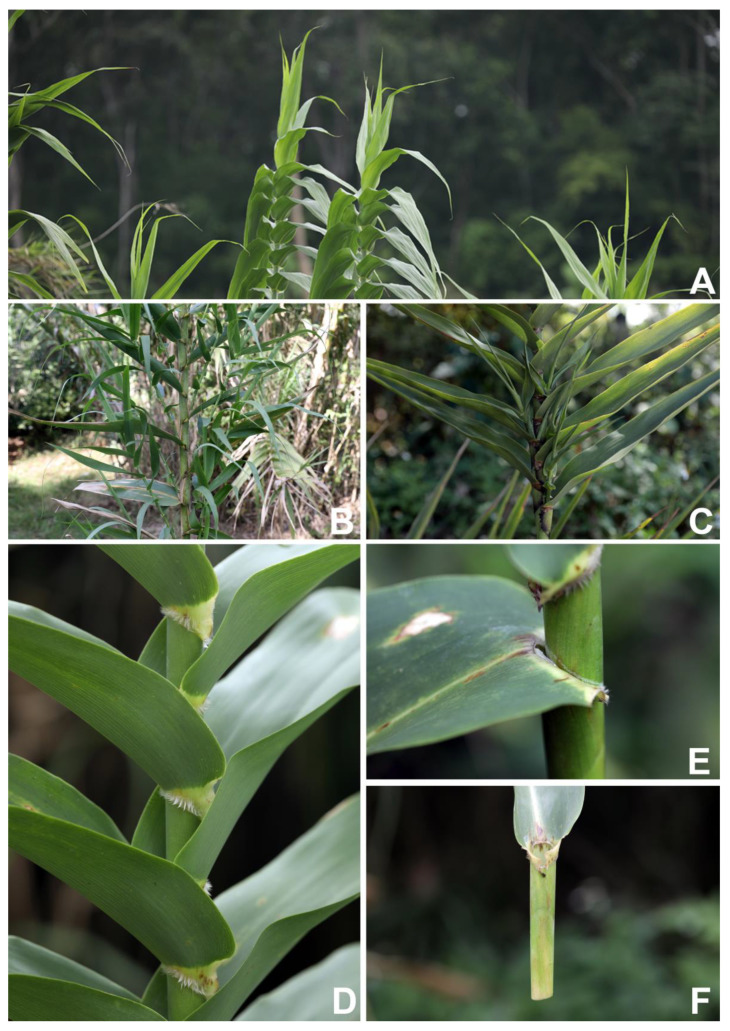
*Arundo donax*. (**A**) distichous cauline leaves; (**B**,**C**) mature culms showing intravaginal branching; (**D**) collars with hairs; (**E**) membranous ligules; and (**F**) cylindrical leaf sheath and auricles. Photos: Chatchai Ngernsaengsaruay.

**Figure 5 plants-12-01850-f005:**
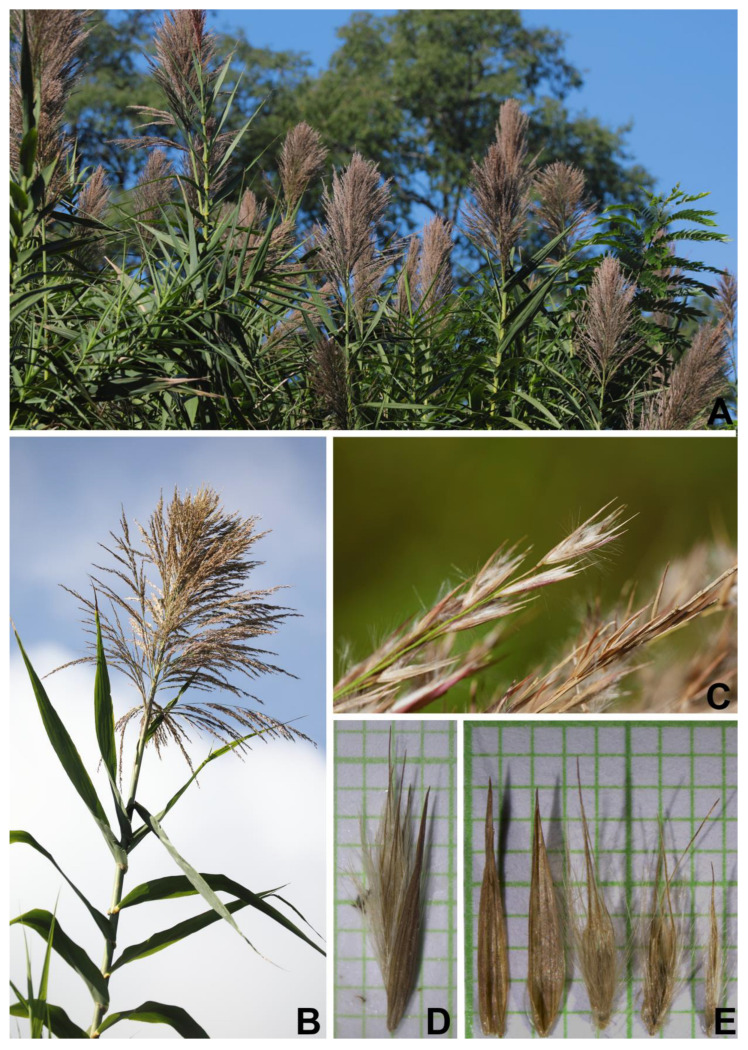
*Arundo donax*. (**A**,**B**) Open, large, plumose, and panicle inflorescences; (**C**) branch of inflorescence with spikelets; (**D**) spikelet; and (**E**) lower glume, upper glume, and florets (from left to right). Photos: (**A**–**C**) Chatchai Ngernsaengsaruay; (**D**,**E**) Pichet Chanton.

**Figure 6 plants-12-01850-f006:**
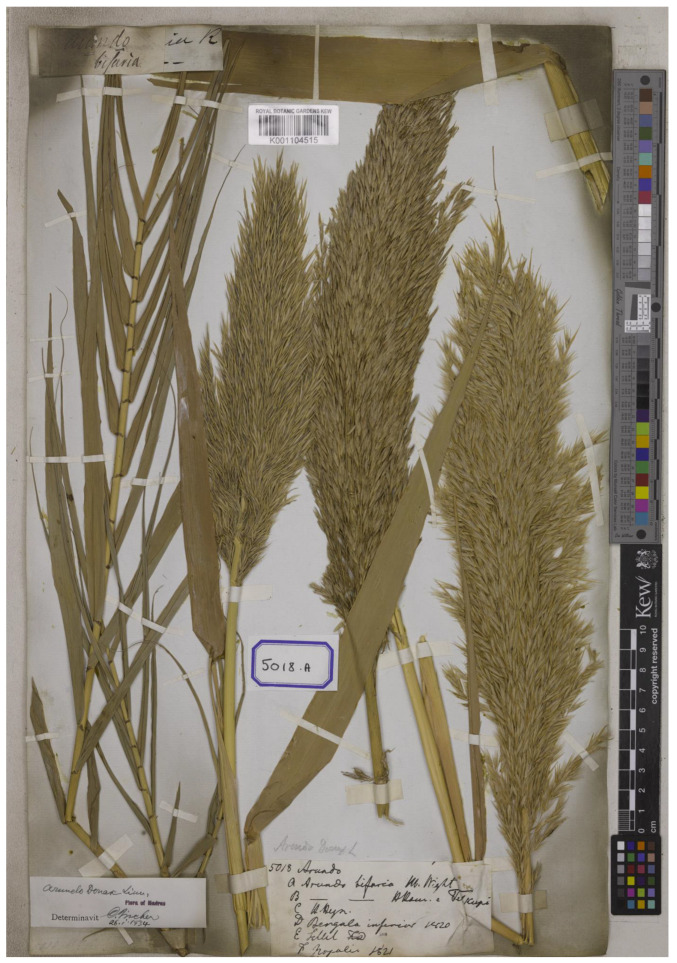
Lectotype of *Arundo bifaria, Wight 1748 (Wight collection, EICH 5018A*) (K-W [K001104515!]) from India, with inflorescences, designated here. Available online: http://specimens.kew.org/herbarium/K001104515 (accessed on 8 February 2023).

**Figure 7 plants-12-01850-f007:**
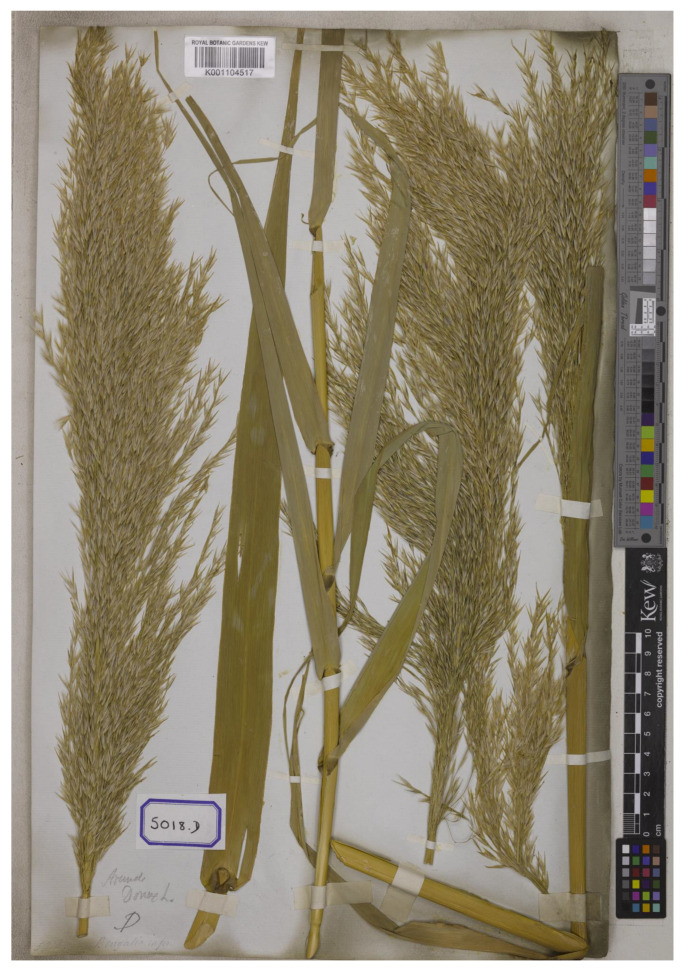
Lectotype of *Arundo bengalensis*, *Wallich Cat. 5018D* (*EICH 5018D*) (K-W [K001104517!]) from Bengal, India, with inflorescences, designated here. Available online: http://specimens.kew.org/herbarium/K001104517 (accessed on 8 February 2023).

*Distribution*. This species is distributed in West and Central Asia to temperate East Asia: Transcaucasus (South Caucasus), Türkiye (Turkey), Cyprus, Egypt (Sinai), Syria, Lebanon, Palestine, Israel, Gulf States (Arab States of the Persian Gulf), Yemen, Kazakhstan, Uzbekistan, Turkmenistan, Tajikistan, Afghanistan, Pakistan, India, Nepal, Bhutan, Bangladesh, Myanmar, Tibet, China, Taiwan, Japan, Vietnam, Laos, Cambodia, and Thailand. This species was very widely introduced into many parts of the world: North and South America, Africa, Europe, Asia, and Australasia (Australia, New Zealand, and some neighbouring islands in the Pacific Ocean). It is an invasive plant species that invades some countries [24], but it is narrowly distributed in its native country or in its range of distribution, especially in Thailand.

*Distribution in Thailand*. Northern: Mae Hong Son [Pang Mapha (Sop Pong and Pang Mapha Subdistr.); Mueang Mae Hong Son (Chong Kham and Pang Mu Subdistr.)], Chiang Mai [Chiang Dao (Chiang Dao, Ping Khong, and Mae Na Subdistr.); Chai Prakan (Pong Tam and Si Dong Yen Subdistr.); Doi Tao (Muet Ka Subdistr.)], Lamphun [Li (Mae Lan Subdistr.)], Phrae (Huai Mae Kham Mi), Nakhon Sawan [Tak Fa (Khao Chai Thong and Tak Fa Subdistr.); Takhli (Takhli Subdistr.)]; Northeastern: Udon Thani [Ban Dung (Ban Dung Subdistr.); Kumphawapi (Huai Koeng Subdistr.)]; Eastern: Chaiyaphum [Thep Sathit (Ban Rai Subdistr.)], Nakhon Ratchasima [Pak Chong (Nong Sarai)]; Southwestern: Kanchanaburi [Thong Pha Phum (Linthin Subdistr.)]; and Central: Chai Nat [Mueang Chai Nat (Hat Tha Sao Subdistr., cultivated); Sapphaya (Taluk Subdistr.)], Bangkok [Suan Luang Rama IX, cultivated] (Figure 8).

*Additional Specimens Examined*. Thailand. Northern: Mae Hong Son [Chong Kham Subdistr., Mueang Mae Hong Son Distr., in disturbed open area along Huai Mae Hong Son, 19°17′27.4″ N, 97°57′04.3″ E, 206 m alt., sterile specimen, 6 May 2022, C. Ngernsaengsaruay et al. *Ad03-06052022* (BKF, QBG); ibid., 19°17′23.0″ N, 97°57′33.0″ E, 230 m alt. (near bus terminal), inflorescences, 24 November 2022 (C. Ngernsaengsaruay et al. own observation, with photos); Ban Sop Pong, Pang Mu Subdistr., Mueang Mae Hong Son Distr., in disturbed open area along Huai Mae Hong Son, 19°17′29.3″ N, 97°57′03.0″ E, 223 m alt. (near the bridge), inflorescences, 22 November 2022, C. Ngernsaengsaruay et al. *Ad07-22112022* (BK, BKF, QBG); ibid., 19°17′39.6″ N, 97°56′54.4″ E, 209 m alt., inflorescences, 22 November 2022, C. Ngernsaengsaruay et al. *Ad08-22112022* (BK, BKF, QBG); Pang Mapha Subdistr., Pang Mapha Distr., in disturbed open area along roadside, near Lang River (near Mae Hong Son Rice Research Centre), 19°32′09.6″ N, 98°13′25.4″ E, 584 m alt., sterile specimen, 6 May 2022, C. Ngernsaengsaruay et al. *Ad04-06052022* (BKF, QBG); ibid., 19°32′09.9″ N, 98°13′24.0″ E, 573 m alt., inflorescences, 23 November 2022, C. Ngernsaengsaruay et al. *Ad09-23112022* (BK, BKF, QBG); ibid., 19°32′10.1″ N, 98°13′28.6″ E, 589 m alt., inflorescences, 23 November 2022, C. Ngernsaengsaruay et al. *Ad10-23112022* (BK, BKF, and QBG); Sop Pong Subdistr., Pang Mapha Distr., in an open area along the Lang River, 19°53′05.6″ N, 98°20′04.8″ E, 560 m alt., inflorescences, 24 November 2022 (C. Ngernsaengsaruay et al. own observation, with photos); ibid., 19°31′05.3″ N, 98°15′57.1″ E, 620 m alt., inflorescences, 24 November 2022 (C. Ngernsaengsaruay et al. own observation, with photos); ibid., 19°32′08.0″ N, 98°12′17.0″ E, 566 m alt., inflorescences, 24 November 2022 (C. Ngernsaengsaruay et al. own observation, with photos); Ban Tha Khrai, Sop Pong Subdistr., Pang Mapha Distr., along the roadside, 500–600 m alt., inflorescences, 15 January 1986, *Y. Pisooksantivatana Y-1774-86* (BK)], Chiang Mai [between Chiang Dao Distr. and Fang Distr., along the riverbank, 350 m alt., sterile specimen, 20 February 1958, *T. Sørensen, K. Larsen and B. Hansen 1398* (E [E00294034]); Ban Mae To, Chiang Dao Subdistr., Chiang Dao Distr., in an open area along Huai Mae To, 19°21′27.7″ N, 98°57′58.0″ E, 384 m alt., sterile specimen, 8 May 2022, C. Ngernsaengsaruay et al. *Ad05-08052022* (BKF, QBG); ibid., 19°20′27.0″ N, 98°58′06.0″ E, 380 m alt., without inflorescence, 16 June 2022 (C. Ngernsaengsaruay et al. own observation, with photos); Ban Sop O, Mae Na Subdistr., Chiang Dao Distr., in an open area along Huai Mae O, 19°18′09.2″ N, 98°57′53.2″ E, 378 m alt., sterile specimen, 8 May 2022, C. Ngernsaengsaruay et al. *Ad06-08052022* (BKF); ibid., 19°18′10.0″ N, 98°57′52.0″ E, 370 m alt., without inflorescence, 16 June 2022 (C. Ngernsaengsaruay et al. own observation, with photos); Ban Mae O Nok, Mae Na Subdistr., Chiang Dao Distr., in an open area along Huai Mae Ying, 19°17′59.0″ N, 98°58′55.0″ E, 390 m alt., without inflorescence, 18 June 2022 (C. Ngernsaengsaruay et al. own observation, with photos); Ban Hua Tho, Chiang Dao Distr., in an open area, 575 m alt., inflorescences, 26 November 2013, *J. F. Maxwell 13-200* (QBG); Ban Hua Tho, Ping Khong Subdistr., Chiang Dao Distr., in an open area along Huai Mae Poi, roadside, 19°34′26.0″ N, 99°05′09.0″ E, 560 m alt., without inflorescence, 17 June 2022 (C. Ngernsaengsaruay et al. own observation, with photos); Chiang Dao Distr., in an open area along Huai Mae Poi, 535 m alt., inflorescences, 3 November 2011, *M. Norsaengsri and N. Tathana 8232* (QBG); Si Dong Yen Subdistr., Chai Prakan Distr., in an open area along the Fang River, 19°38′58.0″ N, 99°09′08.0″ E, 530 m alt., without inflorescence, 17 June 2022 (C. Ngernsaengsaruay et al. own observation, with photos); ibid., 19°38′43.0″ N, 99°08′31.0″ E, 530 m alt. (near Ban Rong Than), without inflorescence, 18 June 2022 (C. Ngernsaengsaruay et al. own observation, with photos); ibid., 19°38′33.0″ N, 99°08′16.0″ E, 530 m alt. (near the bridge), without inflorescence, 18 June 2022 (C. Ngernsaengsaruay et al. own observation, with photos); Ban Rong Than, Si Dong Yen Subdistr., Chai Prakan Distr., in an open area along the Fang River, 19°39′34.0″ N, 99°08′43.0″ E, 570 m alt. (behind Esso petrol station), without inflorescence, 18 June 2022 (C. Ngernsaengsaruay et al. own observation, with photos); Ban Pang Makham Pom, Si Dong Yen Subdistr., Chai Prakan Distr., in an open area along the Fang River, 19°37′47.0″ N, 99°07′40.0″ E, 550 m alt. (near Ban Rong Than School), without inflorescence, 18 June 2022 (C. Ngernsaengsaruay et al. own observation, with photos); Ban Ai, Si Dong Yen Subdistr., Chai Prakan Distr., in an open area along the roadside, 19°41′19.0″ N, 99°08′57.0″ E, 500 m alt., without inflorescence, 17 June 2022 (C. Ngernsaengsaruay et al. own observation, with photos); Ban Pang Khwai, Pong Tam Subdistr., Chai Prakan Distr., in an open area alongside a stream, 19°43′18.0″ N, 99°08′37.0″ E, 500 m alt. (near Doi Wiang Pha National Park), without inflorescence, 17 June 2022 (C. Ngernsaengsaruay et al. own observation, with photos)]; along the banks of Mae Ping (originally “Me Ping” on the label), 300 m alt., inflorescences, 23 October 1912, *A. F. G. Kerr 2750* (K [K000621684]; Ban Muet Ka, Muet Ka Subdistr., Doi Tao Distr. (originally “Ban Mut Ka, Muang Hawt” on the label), 200 m alt., inflorescences, 27 November 1920, *A. F. G. Kerr 4659* (BK, BM [BM000949261], K [K000621688, K000621689]); locality not specified on the label, but “Chiang Mai” on the database of QSBG, inflorescences, 26 October 1995, *M. Norsaengsri 137* (QBG)], Lamphun [Mae Lan Subdistr., Li Distr., in an open area along Huai Mae Hat, 17°41′29.2” N, 98°51′43.3” E, 525 m alt., without inflorescence, 4 February 2023 (Assistant Professor Dr. Nittaya Mianmit’s own observation, with photos)]; Phrae [Huai Mae Kham Mi (originally “Hue Me Kami, Pre” on the label), c. 300 m alt., old inflorescences, 17 February 1921, *A. F. G. Kerr 4854* (BK, BM [BM000949260], K [K000621685, K000621686, K000621687])], Nakhon Sawan [Ban Nong Salao, Khao Chai Thong Subdistr., Tak Fa Distr., along the roadside canal (at the bridge), 15°18′13.2″ N, 100°27′15.4″ E, 80 m alt., old inflorescences, 9 February 2023, C. Ngernsaengsaruay et al. *Ad11-09022023* (BK, QBG); Ban Tak Fa, Tak Fa Subdistr., Tak Fa Distr., along the roadside, 15°20′02.3″ N, 100°29′18.0″ E, 80 m alt., sterile specimen, 9 February 2023, C. Ngernsaengsaruay et al. *Ad12-09022023* (BK, QBG); 284, Ban Dong Poem, Mu 22, Takhli Subdistr., Takhli Distr., 15°18′23.3” N, 100°17′20.2” E, 45 m alt., cultivated as a medicinal plant, without inflorescence, 9 February 2023 (C. Ngernsaengsaruay et al. own observation, with photos)]; Northeastern: Udon Thani [Ban Dung Yai, Ban Dung Subdistr., Ban Dung Distr., 17°42′38.2” N, 103°15′16.3” E, 157 m alt., inflorescences, 4 December 2008, *M. Norsaengsri 4564* (QBG); Ban Sok Khun, Huai Koeng Subdistr., Kumphawapi Distr., by the riverbank, 17°02′43.2” N, 102°56′52.5” E, 190 m alt., inflorescences, 4 December 2008, *M. Norsaengsri 4579* (QBG)]; Eastern: Chaiyaphum [Ban Rai, Ban Rai Subdistr., Thep Sathit Distr., 15°32′87.3″ N, 101°24′77.7″ E, 500 m alt., inflorescences, 6 November 2014, *K. Kertsawang 3241* (BK, QBG)], Nakhon Ratchasima [Nong Sarai Subdistr., Pak Chong Distr., 14°38′39.84″ N, 101°30′43.83″ E, 350 m alt., sterile specimen, 10 March 2023, *C. Ngernsaengsaruay and P. Chanton Ad13-10032023* (BK, BKF, QBG)]; Southwestern: Kanchanaburi [Mae Klong Watershed Research Station, Linthin Subdistr., Thong Pha Phum Distr., in an open area alongside the stream (Professor Dr Dokrak Marod own observation)]; Central: Chai Nat [Ban Taluk, Taluk Subdistr., Sapphaya Distr., along the roadside irrigation canal (Asian Highway, AH2), 15°11′07.4″ N, 100°13′48.4″ E, 20 m alt., without inflorescence, 9 February 2023 (C. Ngernsaengsaruay et al. own observation, with photos); Ban Bang Kan Lueang, Mu 3, Hat Tha Sao Subdistr., Mueang Chai Nat Distr., 15°14′33.0″ N, 100°04′23.8″ E, 20 m alt., cultivated as a medicinal plant, without inflorescence, 9 February 2023 (C. Ngernsaengsaruay et al. own observation, with photos)], Bangkok [Suan Luang Rama IX Park (King Rama IX Park), Prawet Distr., 13°41′26.2” N, 100°39′43.5” E, cultivated inflorescences, 15 December 2021, C. Ngernsaengsaruay et al., *Ad01-15122021* (BK, BKF, and QBG), *Ad02-15122021* (BK, BKF, and QBG); Bangkok (locality not specified), cultivated inflorescences, 24 January 1923, *A. F. G. Kerr s.n.* (BK28545)].

Turkey. Muğla, Ortakent to Dağbelen, *F. Sorger*, *and K. Tan*, *84-11-6* (E [E00294046]).

Palestine. Jericho, *J. E. Dinsmore 2294* (E [E00357022]); Jaffa, *J. E. Dinsmore 4299* (L [L1217358]); Jaffa, *F. Meyers*, *and J. E. Dinsmore B2299* (E [E00357013]).

Isarael. Mt. Carmel, Zikhron Ya’aqov, *D. Zohary*, *and I. Amdursky 482* (E [E00357014], L [L1217357, AMD121716, U1493163, WAG1816577]); Wad el-Kelt, *F. Meyers*, *and J. E. Dinsmore B4299* (E [E00357021]).

Iraq. Darbendikhan, *R. W. Haines s.n.* (E [E00294045]); Dokan, *R. W. Haines s.n.* (E [E00357018]).

Oman. Ayn Sih, Wadi al Ayn, S of Khasab, *M. D. Gallagher*, *and P. R. Sichel* 6399/1 (E [E00294037]).

Iran. Ispahan, *C. P. Bélanger 668* (P [P02657048, P02657049]).

Lebanon. Beyrouth, Beirut *R.*, *Gombault 928* (P [P02656992]); North, Tripoli, *F. Louis s.n.* (P [P02632618, P02632619]).

Tajikistan. Montes meridionales, Sogdiano-transoxanas, *A. Vvedensky 505* (P [P02656979]).

Afghanistan. Unterstes Andarab-Tal, Baghlan, *D. Podlech 12623* (E [E00294038]).

Pakistan. Markan, *S. L. Harris*, *16786* (BM [BM011029608]).

India. Kohima, Assam, *N. L. Bor 17288* (L [L1217387]); Bengale, *C. P. Bélanger s.n.* (P [P02656852, P02656853]); Cachemyr, *V. Jacquemont 1155* (P [P02632632, P02656994, P02656995]); Cachemyr, *V. Jacquemont 1200* (P [P02657000]); Chenab, *C. B. Clarke 23717* (BM [BM011029606]); Chenab, *C. B. Clarke 24107B* (BM [BM000949268]); Darjeeling, *C. B. Clarke 18073* (BM [BM011029615]); Hab. Himal. Bor. Occ., *T. Thomson s.n.* (P [P02657013]); Kanaor, *V. Jacquemont 2102* (P [P02656991, P02657007, P02657008]); Kanaor, *V. Jacquemont s.n.* (P [P02656987]); Drained lake basin, Kashmir, *T. Thomson 4601* (P [P02657012]); Kashmir, *A. P. Young s.n.* (BM [BM011029611]); Kapkol, Kumaon, *R. Strachey, and J. E. Winterbottom s.n.* (BM [BM011029614], P [P02657030]); Maisor and Carnatic, *G. Thomson s.n.* (P [P02657020]); Malabar, Concan, *J. E. Stocks, and Law s.n.* (P [P02657019]); Mont. Khasia, *J. D. Hooker and T. Thomson s.n.* (P [P02657023]); Namjah, *V. Jacquemont 1911* (P [P02656857, P02656989]); Nana Tamul, *J. G. Koenig s.n.* (C [C10016768]); Pondichery, *M. Perrottet 877* (P [P02657026]); Bhuin Kulu, Punjab, *R. E. Cooper and A. K. Bulley 5770* (E [E00576434]); Kothi, Kulu, Punjab, *W. N. Koelz 10213* (BM [BM000949262]); Naggar, Kulu, Punjab, *W. N. Koelz 3048* (E [E00576430], L [L1217391, L1217341]); Punjab, *T. Thomson 58* (P [P02657003]); Punjab, *T. Thomson s.n.* (BM [BM000949267]); Punjab, *T. Thomson s.n.* (P [P02632628, P02632630]); Punjab, *T. Thomson s.n.* (P [P02656983]); Punjab, *T. Thomson s.n.* (L [U1493082]); Tamul, *J. G. Koenig s.n.* (C [C10016769]); Lal gudi, taluk Amoor, Tiruchi Distr., *K. M. Matthew 24658* (L [L1217385]); Kumaon, Uttarakhand, *J. F. Duthie 5102* (BM [BM000949274]); Shillong, *C. B. Clarke 40605B* (BM [BM000949264]); Shillong, *C. B. Clarke 40605C* (BM [BM011029607]); Shillong, *C. B. Clarke 40605E* (E [E00576400]); locality not specified, *Herb. Francis (Buchanan) Hamilton 302* (E [E00576402]); locality not specified, *Herb. Heyne Cat. 5018C* (*EICH 5018 C*) (K [K001104516], P [P02657027]); locality not specified, *J. G. Koenig 88* (K [K000032474]); locality not specified, *J. G. Koenig s.n.* (BM [BM000949266]); locality not specified, *J. G. Koenig s.n.* (C [C10016771]); locality not specified, *W. Roxburgh s.n.* (BM [BM011029609]); locality not specified, *W. Roxburgh s.n.* (BM [BM011029613]).

Nepal. Bhuji Khola, *J. D. A. Stainton, W. R. Sykes, and L. H. J. Williams 9061* (BM [BM000949271], E [E00576417]); Dhankuta, *Sunwar Silose 5* (E [E00294036]); Bhote Kosi en avant de Gongar, Dolkha Distr., *M. A. Farille 81-799* (E [E00190743], P [P00945760]); Entre Loding et Chilanka, Dolkha Distr., *M. A. Farille 81-715* (P [P00945897]); Dolpa Distr., *M. Minaki* et al. *BEWN 9107157* (E [E00241410]); Dolpa Distr., *M. Minaki* et al. *BEWN 9109298* (E [E00241409]); Mugu Karnali Valley, Mangri, *O. Polunin, W. R. Sykes and L. H. J. Williams 3028* (BM [BM011029612], E [E00576416]); Entre Chitre et Sikha, Myagdi Distr., *M. A. Farille 81-271* (E [E00180876], P [P00945727]); below Thulo Syabru, near Pauro Khola, Rasuwa Distr., *D. G. Long* et al. *ENEP 344* (E [E00210089]).

Bhutan. Ganglakha, Phuntsholing Distr., *T. Gyaltsen 10* (E [E00156542]); between Chimakothi and Mathur (Chukka) Bridge, Chukka Distr., *H. J. Noltie* et al. *319* (E [E00166607]). 

Myanmar. Upper Chindwin Distr., *J.F. Rock 802* (P [P02657024, P02657025]); Bilumyo Reserve, Katha State, *J. H. Lace 5530* (E [E00576427]).

China. Guizhou, Songtao Xian, *Sino-American Guizhou Botanical Expedition 2077* (L [L1217384]); Kingchow, *F. A. McClure 10538* (L [L1217388], P [P02656848]); Lin Distr., Kwong Tung Prov., *C. O. Levine 3206* (E [E00576385]); Se Tze Shan, along the Kwangtung border, near Tung Chung village, *W. T. Tsang 22312* (P [P02656981]); Tsekou and Nekou (Haut-Mekong), *R. P. Soulié 1534* (P [P02657035]); Wah Shui Shan, border of Yung-Yuen and Ying Tak Distr., *S. K. Lau 1000* (P [P02656980]); Yunnan, Dêqên Zan Aut. Pref., Zhongdian Co., *Forestry Commission, RBGE Dêqên Expedition (1995) 459* (E [E00051204]); Long-Ky, Yunnan, *E. E. Maire s.n.* (E [E00576386, E00576389]); Lushui, Huangcaoping, Yunnan, *H. Li* et al. *10259* (E [E00226753]); Lushui Xian, Yunnan, *H. Li* et al. *10360* (E [E00248124]); Nujiang Lisu Aut. Pref., Fugong Co., Yunnan, *Gaoligong Shan Biotic Survey Expedition (1996) 7936* (E [E00161195]); Nujiang Lisu Aut. Pref., Gongshan Co., Yunnan, *Gaoligong Shan Biotic Survey Expedition (1997) 9030* (E [E00107622]); Plaine de Tchao-Tong, Yunnan, *R. P. Maire s.n.* (P [P02657016]); Yunnan, *M. l’Abbé Delavay s.n.* (P [P02657017, P02657018]); Yunnan, *M. l’Abbé Delavay 2976* (P [P02657033]); Yunnan, *M. l’Abbé Delavay 2978* (P [P02657034]).

Taiwan. Bankinsing, *A. Faurie 137* (P [P02657037]); Tawu, Taitung County, *C. E. Devol 7209* (L [L1217337]).

Japan. Awaji, Honshiu, *G. Murata 15831* (L [U1493165)]); Bonin Islands, *Anonymous s.n.* (E [E00576390]); Fotomi, Anai, *Anonymous s.n.* (E [E00576393]); Fotomi, Hamamatsu, *Anonymous s.n.* (E [E00576394]); Sagami, Hondo, Ashigarashimo-gun, Yugawara, *T. Makino 6868* (E [E00576392]).

Vietnam. Cho Ganh, *M. Pételot 1314* (P [P02657047, P02656854]); Doug-Pang, in the bottom of the ravine, *B. Balansa 349* (P [P02656982]); Phuc Nhae, *Le R. P. Bon 1882* (P [P02656850]); Tonkin, *B. Balansa 4713* (P [P02656849]).

Habitat and ecology. In Thailand, it is mostly found in wet habitats such as open areas along the riverbanks, streams, seasonally flooded areas, and irrigation canals, but also in drier habitats along the roadsides at elevations of 20–620 m above mean sea level. (Figure 9). In the lower Himalaya, from Kashmir to Nepal, it occurs up to c. 1070 m alt., and from Punjab to Sylhet, in the Naga Hills, it occurs at 1520–2440 m alt. [10]. Bor stated that it will grow in dry habitats when established, but it prefers plenty of moisture [11]. In Southeast Asia, it occurs chiefly in low, wet sites such as riverbanks and seasonally flooded places, but also in drier habitats at altitudes less than 1800 m [16]. In Japan and its neighbouring regions, it is occasionally found on sandy dunes along the coast [18].

IUCN Conservation Status: Least Concern (LC) [25]. This species is widely distributed from West and Central Asia to temperate East Asia and has been widely introduced into many parts of the world. The Global Invasive Species Database stated that dense populations of *Arundo donax* affect riversides and stream channels, compete with and displace native plants, interfere with flood control, and are extremely flammable, increasing the likelihood and intensity of fires. It may establish an invasive plant-fire regime as it both causes fires and recovers from them 3–4 times faster than native plants. It is also known to displace and reduce habitat for native species. Long ‘lag times’ between introduction and development of negative impacts are documented in some invasive species; the development of *A. donax* as a serious problem in California may have taken more than 400 years. The opportunity to control this weed before it becomes a problem should be taken, as once established, it becomes difficult to control [24]. It is appropriate to consider its status as LC.

*Phenology.* Flowering from October to December; fruiting from November to December.

*Etymology. Arundo donax* was named by Carl Linnaeus (1707–1778), a Swedish botanist, physician, and zoologist [26]. The generic name “*Arundo*” comes from the old Latin name for a reed. The specific epithet *“donax”* from the old Greek name for a reed [27,28].

*Vernacular Name*: O (อ้อ) (General); O luang (อ้อหลวง) (Northern); O yai (อ้อใหญ่) (Central) (Thai); Tam pa-dong (ตำปะดง) (Chaobon-Chaiyaphum); Bamboo reed, Colorado river reed, Cow cane, Donax cane, Elephant grass, Giant cane, Giant reed, Giant reed grass, Spanish cane, Spanish reed, Wild cane (English); Narkato (Nepali); and Danchiku (Japanese).

*Uses.* In the northern region of Thailand, the woody culms of *Arundo donax* are locally used in light construction and are used in building rough fences, walls, and benches. When they are crushed and sun-dried, they can be woven into rough mats to be used as walls. They are also used to make giant reed tubes (giant reed cylinders) with sand, uncooked rice, or water contained inside the hollow culm internodes used in the traditional Lanna rites of “Suep Chata” (the prolongation rituals of human life) (based on the observations of C. Ngernsaengsaruay et al. and interviews conducted in Mae Hong Son, Chiang Mai, and Lamphun Provinces). *A. donax* is also grown in the garden of Suan Luang Rama IX Park (King Rama IX Park) as an ornamental grass to provide botanical education to the people (C. Ngernsaengsaruay et al. own observations and interviews). In the northern and central regions of Nakhon Sawan and Chai Nat Provinces, respectively, it is cultivated as a medicinal grass. The young culms and young leaves are cut and boiled in water for bathing and used as a treatment for an itching rash (called “Pa dong”) by the local people. (C. Ngernsaengsaruay et al. own observations and interviews). In the Chaiyaphum province, located in the eastern part of Thailand, the whole plant of this species is boiled in water for bathing or drinking and is used in the treatment of eczema (dry, itchy skin) by the Kyah Kur (from the specimen *K. Kertsawang 3241*). *A. donax* has long been used in the European region; the dried culm internodes are used in musical instruments as reeds for clarinet, oboe, and bassoon [9]. Dr. Nattapon Banjatammanon, a lecturer and clarinet player from the Department of Music, Faculty of Humanities, Kasetsart University, has been purchasing dried culms of giant reeds from France for making clarinet reeds himself (Figure 10).

Previous studies reported that the culms of *Arundo donax* are a useful source of cane for light construction and for making woodwind instrument reeds [6,7]. The culms are used for making clarinet reeds and organ pipes [24,29] and are extensively cut to make mats, trays, and baskets, with the Romans using the culms to make pens and sometimes paper [17]. The dried, crushed culms can be woven into rough mats to be used in making walls and roofs. The hollow culms are also used to make herdsmen’s pipes. In Texas, this grass has been planted to protect against wind erosion [11] and has been extensively used by the highway department for erosion control and beautification along the culverts and bridges [30]. It has long been grown as a windbreak in horticultural plantings, but more recently it has been brought into cultivation for reed production [8]. The culms are also used to make sticks and fishing rods [15]. The large, thick, and fluffy flower plumes (inflorescences) are used in floral arrangements [24].

*Notes.* The local name “Tam pa-dong” is Nyah Kur language, also called Chaobon or Chaobon-Chaiyaphum (from the specimen *K. Kertsawang 3241*). The Napali name “Narkato” from the specimen *Sunwar Silose 5* at E [E00294036].

The names “Ban Sop O”, “Huai Mae O”, “Ban Mae O Nok”, and “Ban Mae O Nai” are located in Mae Na Subdistrict., Chiang Dao District, Chiang Mai Province, the northern region of Thailand. They are geographical names (place names or toponyms) that are applied to topographical features and settled places in *Arundo donax*. The words “Ban” means the village, “Sop” means the junction of rivers, “Huai” refers to the stream, and “O” is the local name of *A. donax*.

### 2.2. Culm Internode and Leaf Blade Anatomy

#### 2.2.1. Culm Internode Anatomy

The culm internodes of *Arundo donax* are circular with a wide hollow central pith (central pith cavity), leaving a cylindrical area of tissue (solid tissue). A monolayer of epidermis and several layers of cortex that are both small cells. There is a continuous sclerenchyma ring (also called a zone of sclerenchymatous cells or a fibre band) between the cortex and ground tissue close to the periphery. The outermost small vascular bundles are embedded in a continuous sclerenchyma ring. Numerous larger vascular bundles are scattered in the ground tissue, and the parenchyma cells have significantly lignified cell walls. Vascular bundles are composed of phloem and xylem (conducting tissue) and are surrounded in a continuous sclerenchymatous bundle sheath (also called a bundle sheath ring or fibre ring). The size of the vascular bundles (including sclerenchyma rings) increased gradually with the distance inward from the epidermis (Figure 11). Measurements of the vascular bundles, vessels, and parenchyma cells are shown in Table 3.

According to Kawasaki et al. [9], it turned out that the acoustic quality of a reed is mainly ascribed to the shape and configuration of vascular bundles and the size of parenchyma cells. A reed where vascular bundles with continuous bundle sheath rings are homogeneously distributed with a higher proportion among a softer network of small parenchyma cells enables musical performance, and Veselack [31] found that clarinet reed quality was related to the size of the parenchyma cells, the size of vascular bundles and the associated bundle sheath ring, and the number of twisted and broken bundle sheath rings. The results of Kawasaki et al. [9] are consistent with Kolesik et al. [8]. Furthermore, from our study, we found the vascular bundles are surrounded in a continuous sclerenchymatous bundle sheath, homogenously distributed, and enclosed by small parenchyma cells. These are good characteristics of culm internodes as the material for clarinet reeds made from *Arundo donax*, in agreement with Veselack [31], Kolesik et al. [8], and Kawasaki et al. [9].

#### 2.2.2. Leaf Blade Anatomy

*Transverse section of leaf blades*: The outline of the leaf blades in the transverse section is flat. Bulliform cells are present on the adaxial surface in discrete, regular groups, each cell pyriform, and combine with colourless mesophyll cells to form narrow groups penetrating the mesophyll. All the veins appear parallel to each other. The vascular system consists of three orders of collateral vascular bundles (variously sized: large-sized, middle-sized, and small-sized); the median vascular bundle (first-order vascular bundle) is larger than the lateral vascular bundles (second-order vascular bundles and third-order vascular bundles). Vascular bundles are composed of phloem and xylem (vascular tissue) and are enclosed in two layers of bundle sheath cells: outer bundle sheath cells and inner bundle sheath cells. The outer bundle sheath cells have thin walls and are colourless (parenchyma cells), and the inner bundle sheath cells have thick walls (sclerenchyma cells). The chloroplasts in the transverse section of the leaf blades are found only in the mesophyll cells but are absent in the bundle sheath cells, which indicates that *Arundo donax* is a C3 grass, in agreement with Watson and Dallwitz [7] (Figure 12).

*Leaf blade epidermis*: The leaf blade epidermis of *Arundo donax* consists of long cells alternating with short cells arranged in the costal and intercostal zones on both surfaces. The long costal cells are elongated and rectangular. The intercostal long cells are irregularly elongated with sinuous anticlinal walls (arranged in vertical rows and having sinuous radial walls), and the intercostal short cells (silica cells) have silica bodies. The styloid crystals can be found in the leaf blade epidermal cells. The leaves are amphistomatic (have stomata on both surfaces) and are confined to the intercostal zones. The stomata are typically paracytic, with two lateral subsidiary cells placed parallel to the guard cells. The subsidiary cells are dome-shaped, and the guard cells are dumbbell-shaped. The size of stomata (including subsidiary cells) is 23.86–37.12 (30.67 ± 3.24) × 16.10–28.31 (21.38 ± 2.21) µm. The stomatal density is higher on the abaxial surface [450–839/mm^2^ (606.83 ± 72.71)] relative to the adaxial surface [286–587/mm^2^ (441.27 ± 50.72)]. Bulliform cell rows can be distinguished in the intercostal zone of the adaxial surface (Figure 13).

### 2.3. Palynology

The pollen grains of *Arundo donax* are monads, heteropolar, and radially symmetrical. The shape of pollen is spheroidal or subspheroidal [P/E ratio = 0.89–1.16 (1.02 ± 0.07)], the polar axis diameter is 25.38–38.32 (29.98 ± 3.49) µm, and the equatorial axis diameter is 24.95–38.13 (29.52 ± 3.79) µm, which are medium-sized. The pollen aperture is monoporate, and the pore is circular, 3.86–7.25 (5.54 ± 0.92) µm in diameter, with a faint annulus (an area of the exine surrounding a pore). The exine thickness is 1.08–1.97 (1.53 ± 0.27) µm, and the sculpturing is granular (Figure 14 and Figure 15).

## 3. Discussion

In its natural habitats in Pakistan, *Arundo donax* can reach 5 m tall [17], in China, it grows 2–6 m tall [20], and in Japan and its neighbouring regions, it grows 2–5(–8) m tall [18]. This species is naturalised in warm temperate and tropical parts of the Americas; it grows up to 8 m tall [32]. It was introduced and cultivated in Peninsular Malaysia and Singapore; it grows 1.5–3 m tall [14,19], and in Java, it grows 1.5–4 m tall [13]. In addition, from our field observations and examination of specimens in Thailand, it grows 3–8.2 m tall. Plant growth and geographical distribution are affected by environmental factors.

According to Backer and Bakhuizen van den Brink [13] and Watson and Dallwitz [7], the branches of the genus *Arundo* are extravaginal; however, from our observations, we found the branches of *Arundo donax* are intravaginal (Figure 4B,C).

According to previous studies, the sizes of leaf blades are c. 40 × 1.5–4 cm [19], 30–60 × 2–5 cm [20], 30–60 × 2.5–5 cm [17], 15–75 × 0.8–4.5 cm [13], up to 75 × 4 cm [14], 50–70(–90) × 2–6 cm [18], and up to 1 m × 6 cm [32]. Furthermore, from our study, we found the size of leaf blade has a range of 24–73.2 × 2–11.7 cm, sometimes wider than previous studies.

According to previous studies, the lengths of inflorescences are 30–50 cm [12], 30–50 cm long or more [14], 30–60 cm [10,17,20], up to 60 cm [32], 30–70 cm [18], and 30–75 cm [13]. In addition, we found the length of inflorescences to have a range of 70–165 cm (including the peduncle) and 42–80 cm (excluding the peduncle) in this study, sometimes longer than previous studies. The lengths of primary branches of inflorescences are 10–25 cm [20] and 20–30 cm (longest) [19], but we found lengths of (5–)10–43 cm in this study, sometimes longer than previous studies.

The spikelets of this species have 3–4 florets [10,13,19], 3–4 florets or more [15], 2–5 florets [20], 3–5 florets [12,18], and 4–5 florets [32], but from our study, we found 2–3(–4) florets.

The glumes of this species have three veins [10,12,18], but we found three or five veins in this study, in agreement with Backer and Bakhuizen van den Brink [13], Hsu [15], Cope [17], and Chen et al. [20].

The lemmas of this species have 3–5 veins, rarely 7 veins [18], 3–7 veins [20,32], 5 veins [15], 5–9 veins [14], and 8 veins [13], but from our study, we found 3 or 5 veins in conformity with Lee [12] and Cope [17].

The fruit of *Arundo* is a caryopsis and oblong [10,15], but we have not seen it in this study. A discussion about the anatomy is mentioned in the result.

According to previous studies, the sizes of pollens are 12 × 12 µm [33] and in the range of 29–40 µm in diameter [34]. Furthermore, from our study, we found the pollen diameter in the equatorial axis to be in the range of 24.95–38.13 (29.52 ± 3.79) µm, consistent with Trigo and Fernández [34].

According to Sanghi and Sarna [33], the exine sculpturing is psilate; however, from our observations, we found it to be granular.

## 4. Materials and Methods

Plant specimens of *Arundo donax* were observed and collected in the Northern (Mae Hong Son, Chiang Mai, Lamphun, and Nakhon Sawan), Eastern (Nakhon Ratchasima), and Central (Chai Nat and Bangkok) regions of Thailand (Table A1, Figure A1). Herbarium specimens deposited in BK, BKF, QBG, and those included in the digital herbarium databases of BM, C, E, JSTOR, K, K-W, L, and P were examined by consulting the taxonomic literature (acronyms follow the study by Thiers [35]). The herbarium accession number can be seen on the specimens examined. The taxonomic history of this species was compiled using the taxonomic literature and online databases [5,36]. The morphological characters, distribution, habitat, phenology, and uses were described from our observations and interviews during field work and from label information on the specimens examined. The vernacular names were compiled from the specimens examined and the literature [18,24,37,38].

The preparation of plant samples was for anatomical observation. Transverse and longitudinal sections of the culm internodes, as well as transverse sections of the leaf blades, were through the midribs. The culm internode samples were sectioned with a sliding microtome at 16–20 µm thickness, stained with safranin and fast green, and mounted in DePex mounting media. The leaf samples were dehydrated in an increasing ethanol concentration series of 30%, 50%, 70%, 95%, and absolute ethanol, embedded in paraffin, sectioned with a rotary microtome at 16–20 µm thickness with Haupt’s adhesive affixing paraffin sections to slides, stained in safranin and fast green, cleared with xylene, and mounted in DePeX mounting media. Leaf epidermal preparations were made by peeling and mounting on slides. The anatomical characteristics were investigated and recorded photographically with an Olympus BX53 microscope and an Olympus DP74 microscope digital camera at the Department of Botany, Faculty of Science, Kasetsart University (KU). The anatomy terminologies follow the study by Metcalfe [39].

The samples of pollen grains were taken from the herbarium specimen collected from Suan Luang Rama IX Park, Bangkok (C. Ngernsaengsaruay et al., *Ad01-15122021*). They were examined and recorded photographically with an Olympus BX53 microscope and an Olympus DP74 microscope digital camera. Materials were prepared for scanning electron microscopy (SEM) at the Scientific Equipment Centre, Faculty of Science, KU by mounting pollen grains on stubs using double-sided sellotape, sputter-coated with gold, and examined by a FEI Quanta 450 SEM (Hillsboro, OR, USA) at 15.00 KV. The characteristics of pollen grains (size, shape, symmetry, aperture, exine thickness, and sculpturing) were examined and measured. The pollen morphology terminologies follow Punt et al. [40].

## 5. Conclusions

The morphology, taxonomy, culm internode and leaf blade anatomy, and palynology of *Arundo donax* are reported. *Arundo* is related to *Phragmites* but differs in having its lemma with long white hairs outside below the middle part (vs. glabrous); rachilla glabrous (vs. with long white hairs); glumes subequal, as long as spikelet (vs. unequal, shorter than spikelet); and the lowest floret bisexual (vs. male or sterile). Two names in *Arundo* are lectotypified: *A. bifaria* and *A. bengalensis*, which are synonyms of *A. donax*. It is mostly found in wet habitats such as open areas along riverbanks and streams up to elevations of 620 m above mean sea level, especially in the northern region of Thailand. The culms are locally used for light construction, building rough fences, walls, and benches, and are also used in the traditional Lanna rituals of “Suep Chata”. *A. donax* is also cultivated as a medicinal grass and is used in the treatment of itching rashes. It is also cultivated in the garden of Suan Luang Rama IX as an ornamental grass to provide botanical education for people. Dr. Nattapon Banjatammanon, a lecturer and clarinet player from the Department of Music, Faculty of Humanities, Kasetsart University, and a co-author in this paper, has been purchasing dried culms of *A. donax* from France for making clarinet reeds himself. The anatomical characteristics of the culm internodes affect the musical performance of clarinet reeds made from *A. donax*. The chloroplasts in the transverse section of the leaf blades are not found in the bundle sheath cells, which indicates that it is a C3 grass. Generally, the pollen grains in Poaceae are spheroidal with a single pore (monoporate) surrounded by an annulus, and the exine sculpturing is granular.

*Arundo donax* can be promoted as the next cash crop due to its fast growth and diverse usage. Reeds for woodwind musical instruments are a possible market. Moreover, traditional medicines and light construction also have economic benefits. Plantations of *A. donax* would be under consideration in other countries where it is naturally distributed, including Thailand as well.

## Figures and Tables

**Figure 8 plants-12-01850-f008:**
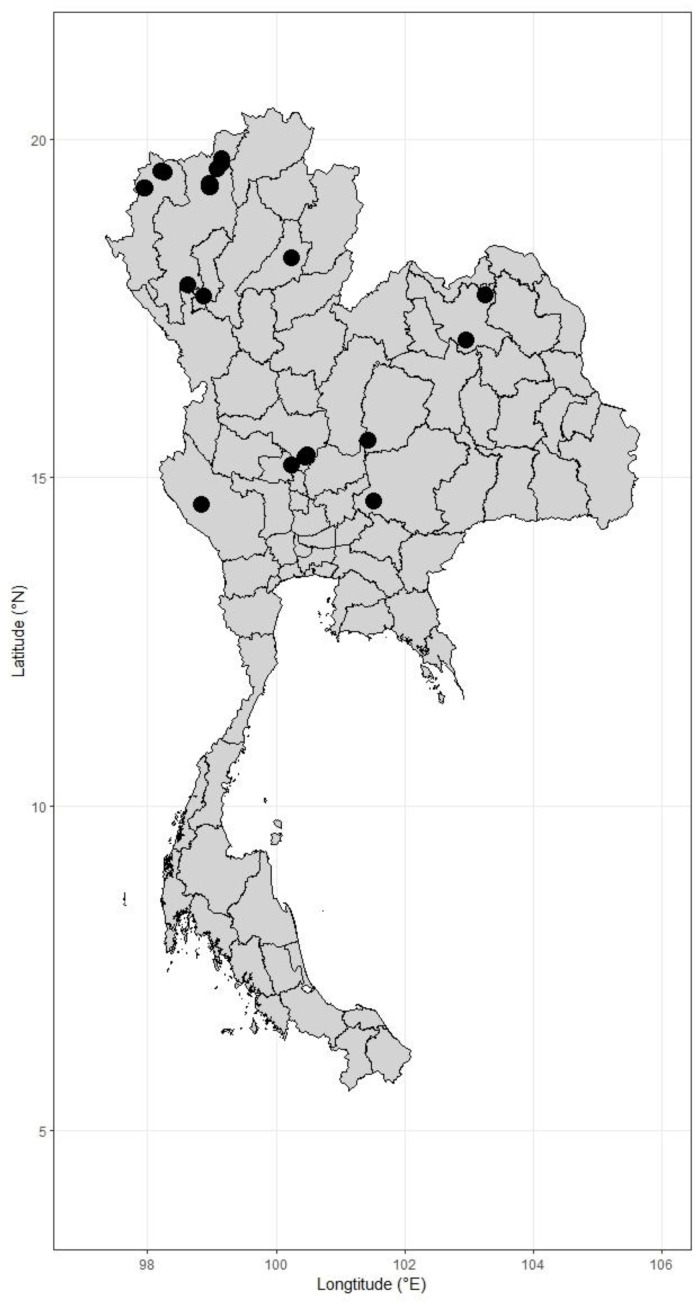
Distribution of *Arundo donax* in Thailand. It is distributed in northern, northeastern, and southwestern.

**Figure 9 plants-12-01850-f009:**
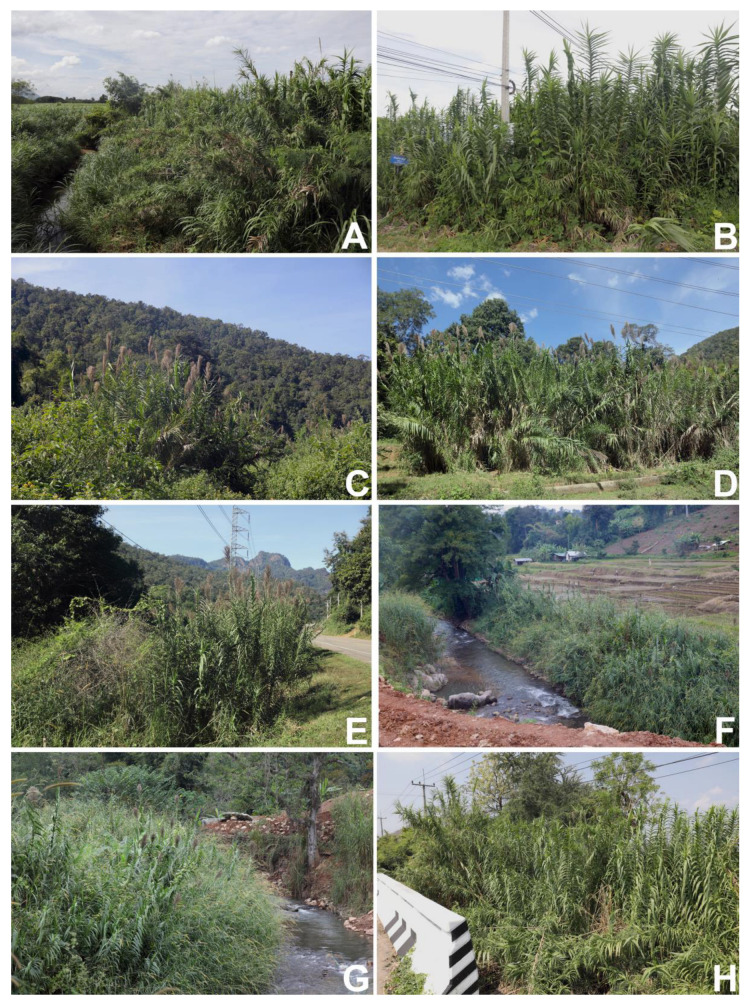
Habitat of *Arundo donax* in Thailand. (**A**) in an open area along Huai Mae To, Ban Mae To, Chiang Dao Subdistrict, Chiang Dao District, and Chiang Mai Province; (**B**) in an open area along the roadside, Ban Ai, Si Dong Yen Subdistrict, Chai Prakan District, and Chiang Mai Province; (**C**–**E**) in a disturbed open area along the roadside, near Lang River, Pang Mapha Subdistrict, Pang Mapha District, and Mae Hong Son Province; (**F**,**G**) in an open area along the Lang River, Sop Pong Subdistrict, Pang Mapha District, and Hong Son Province; and (**H**) along a roadside canal, Ban Nong Salao, Khao Chai Thong Subdistrict, Tak Fa District, and Nakhon Sawan Province. Photos: Chatchai Ngernsaengsaruay.

**Figure 10 plants-12-01850-f010:**
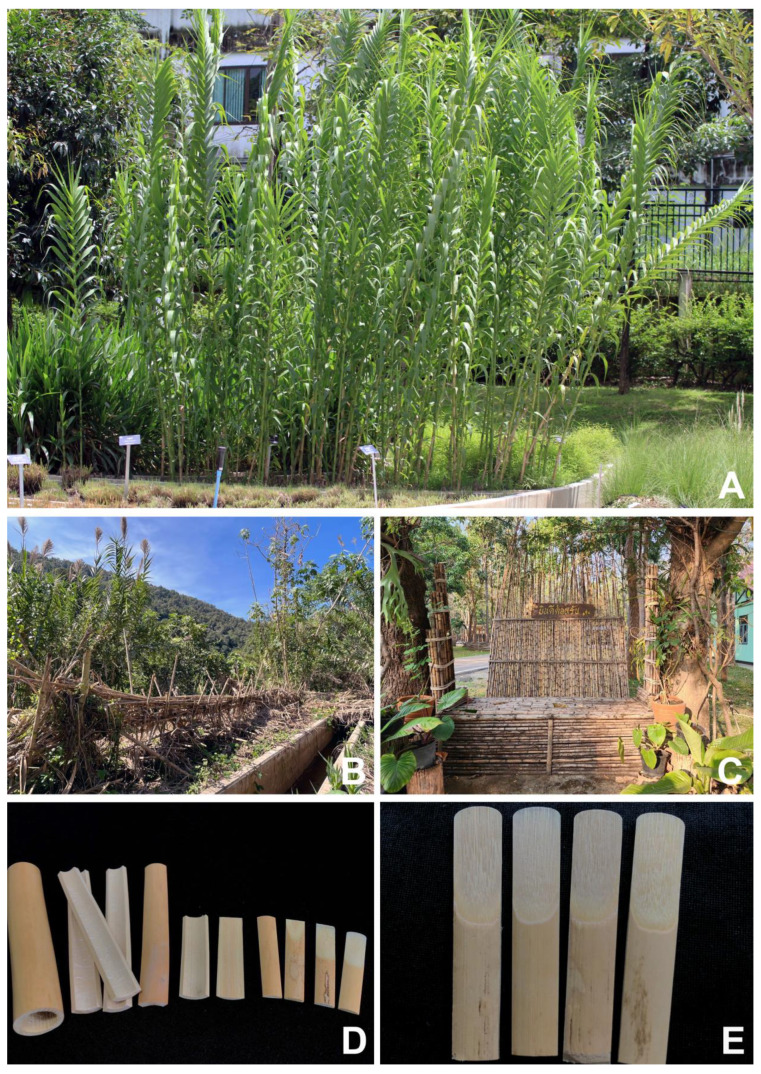
Uses of *Arundo donax* in Thailand. (**A**) cultivated as an ornamental grass to provide botanical education to the people; (**B**) used in building rough fences; (**C**) used in building rough walls and benches; and (**D**,**E**) used to make clarinet reeds. Photos: (**A**,**B**) Chatchai Ngernsaengsaruay; (**C**) Nittaya Mianmit; and (**D**,**E**) Piyawan Yimlamai.

**Figure 11 plants-12-01850-f011:**
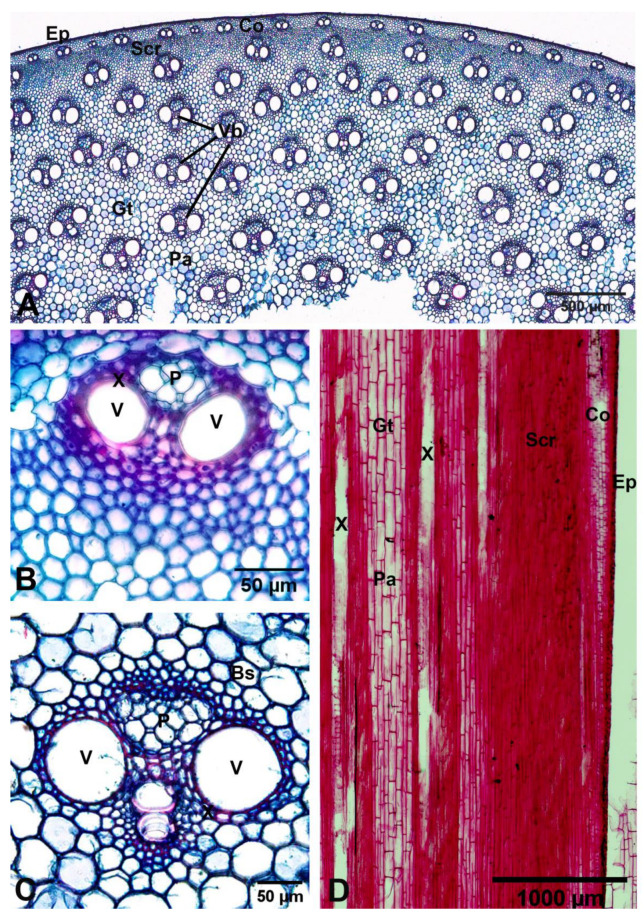
Culm internode anatomical characteristics of *Arundo donax*. (**A**–**C**) transverse sections; (**A**) low-magnification image; (**B**) high-magnification image, showing the outermost small vascular bundle; (**C**) high-magnification image, showing the vascular bundle in the ground tissue; and (**D**) longitudinal section. Bs: bundle sheath, Co: cortex, Ep: epidermis, Gt: Ground tissue, P: phloem, Pa: parenchyma cell, Scr: sclerenchyma ring, V: vessel, Vb: vascular bundle, X: xylem. Photos: Pichet Chanton and Thirawat Thaepthup.

**Figure 12 plants-12-01850-f012:**
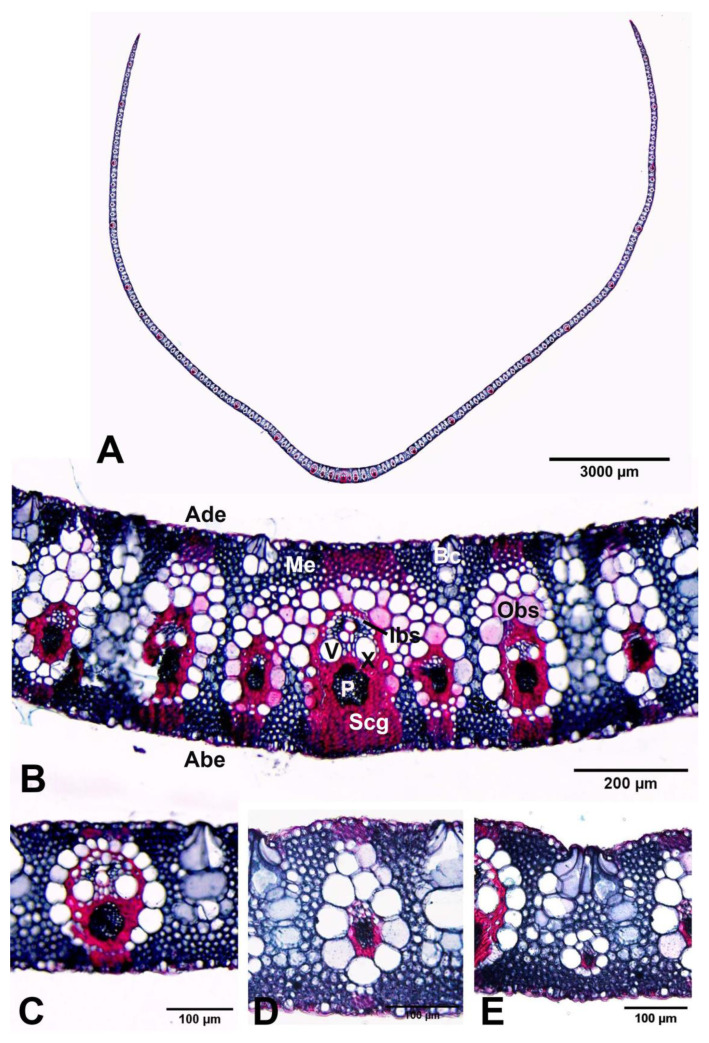
Transverse sections of leaf blades of *Arundo donax*. (**A**) whole leaf blade; (**B**) median and lateral vascular bundles; (**C**,**D**) second-order vascular bundles; and (**E**) third-order vascular bundle. Abe: abaxial epidermis, Ade: adaxial epidermis, Bc: bulliform cells, Ibs: inner bundle sheath, Me: mesophyll, Obs: outer bundle sheath, Scg: sclerenchyma girder. Photos: Pichet Chanton and Thirawat Thaepthup.

**Figure 13 plants-12-01850-f013:**
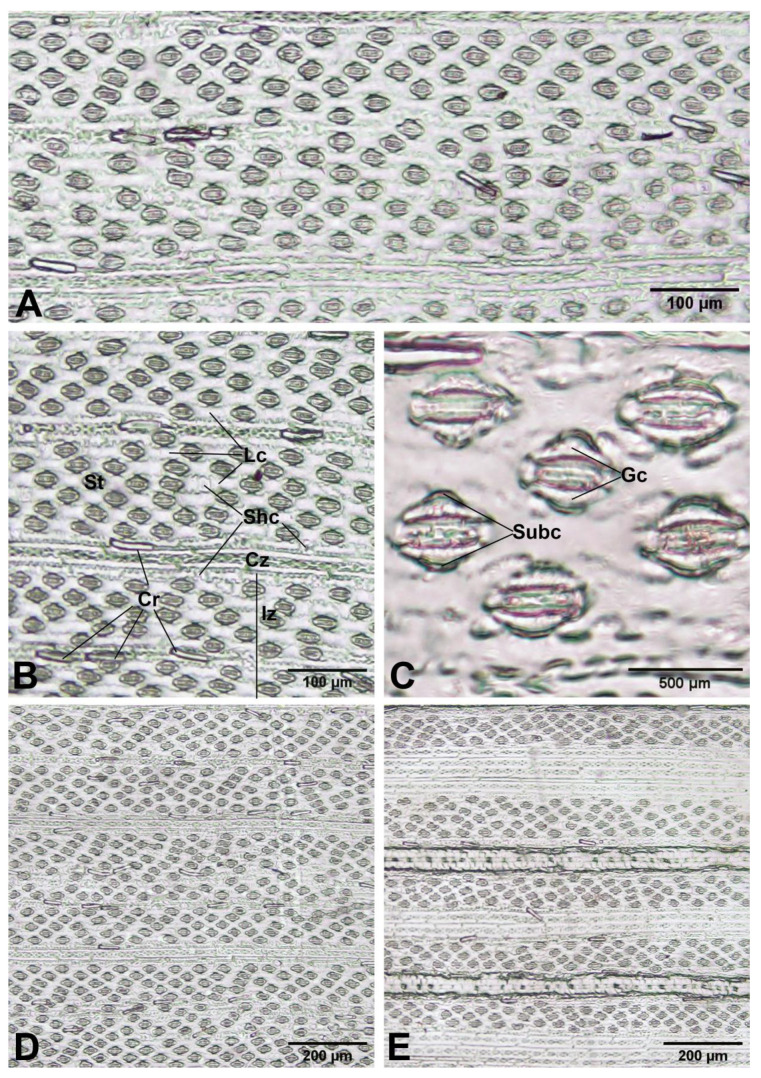
Leaf blade epidermis of *Arundo donax*. (**A**–**D**) abaxial surfaces; and (**E**) adaxial surface. Cr: styloid crystals, Cz: costal zone, Gc: guard cells, Iz: intercostal zone, Lc: long cells, Shc: short cells, St: stoma, Suc: subsidiary cells. Photos: Pichet Chanton and Thirawat Thaepthup.

**Figure 14 plants-12-01850-f014:**
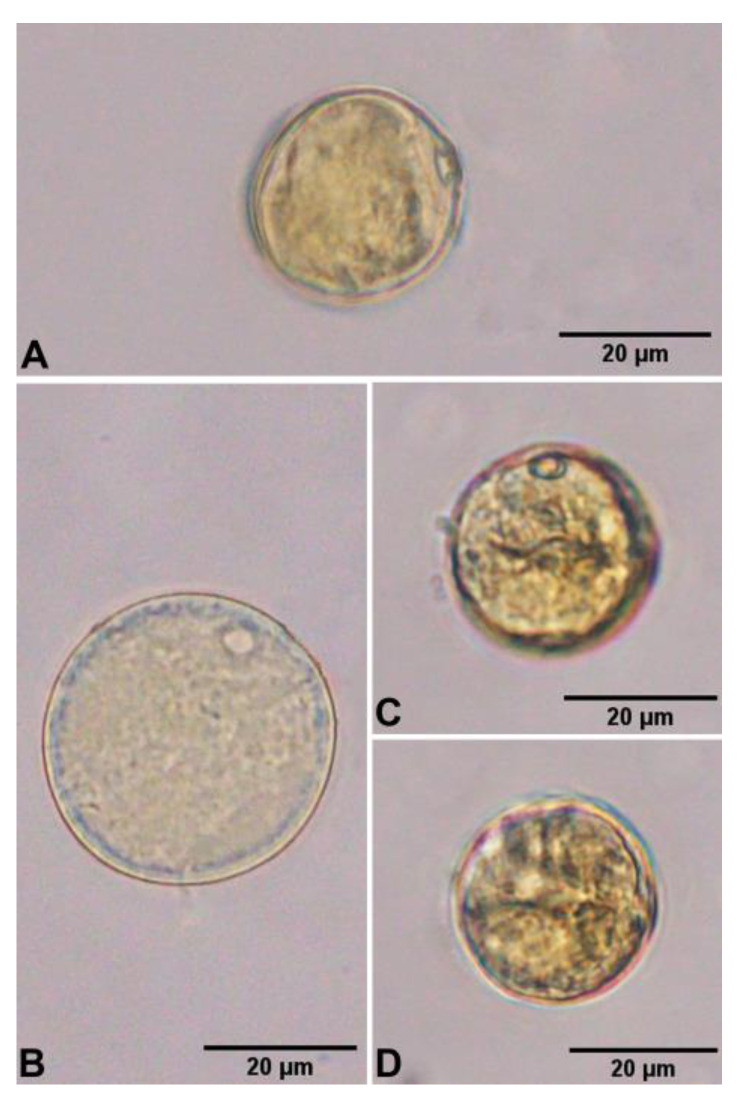
(**A**–**D**) LM micrographs of pollen grains of *Arundo donax*. Photos: Pichet Chanton.

**Figure 15 plants-12-01850-f015:**
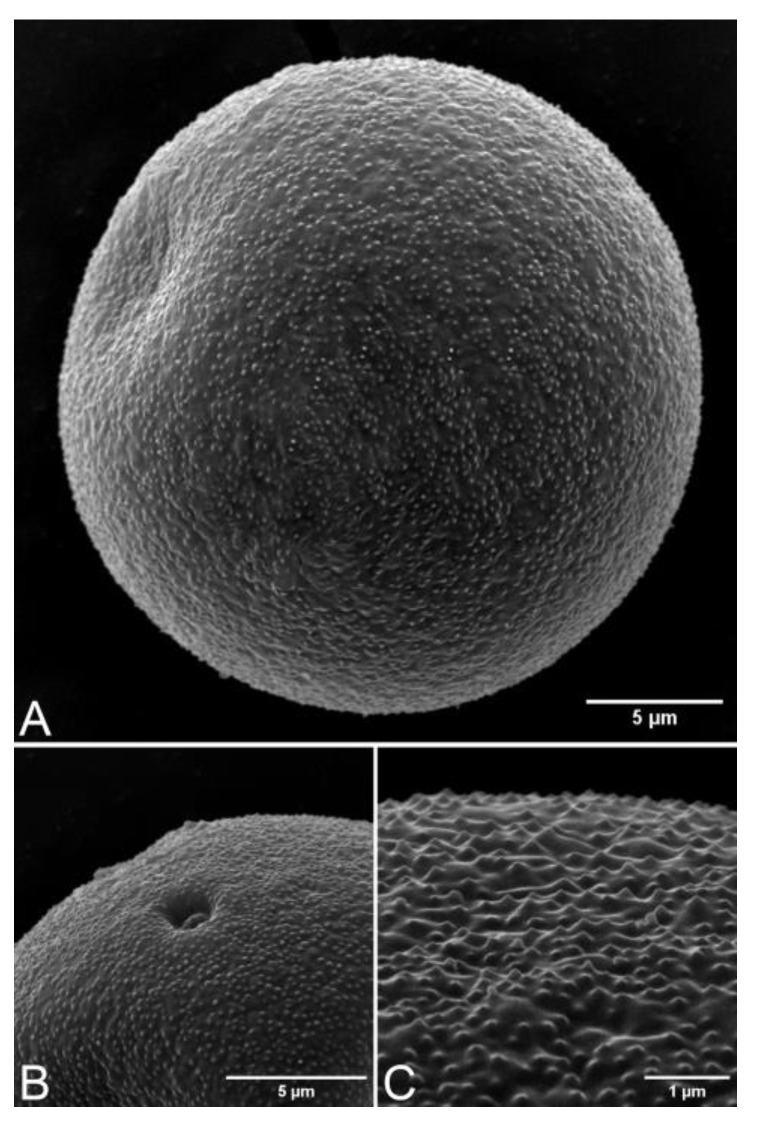
(**A**–**C**) SEM micrographs of pollen grains of *Arundo donax*.

**Table 1 plants-12-01850-t001:** A comparison of morphological characteristics of *Arundo* and related genera, *Phragmites* and *Elytrophorus*, in Arundinoideae.

Characteristics	*Arundo* Tribe Arundineae	*Phragmites* Tribe Molinieae	*Elytrophorus* Tribe Crinipedeae
Habit	Large, tall, rhizomatous perennial reed	Large, tall, rhizomatous perennial reed	Annual grass, caespitose (tufted), without rhizomes, up to 60 cm tall
Culms (Stems)	Herbaceous (soft, non-woody tissue) when young, becoming woody (wood-like) when mature; culm internodes are hollow	Herbaceous when young, becoming woody when mature; culm internodes are hollow	Herbaceous; culm internodes are solid
Leaves	Cauline; leaf blades broad, up to 11.7 cm wide, disarticulating from the leaf sheaths (deciduous)	Cauline; leaf blades broad, up to c. 5 cm wide, disarticulating from the leaf sheaths (deciduous)	Basal; leaf blades narrow, non-disarticulating from the leaf sheaths (persistent)
Ligules	Membranous with a minutely ciliolate margin	Very short membrane with a long ciliate margin	Unfringed membrane, or a fringed membrane
Inflorescences	Open, large, plumose panicle	Open, large, plumose panicle	Contracted panicles of dense globular spikelet clusters borne at intervals along a central axis and sometimes confluent to form a cylindrical shape
Spikelets	Solitary, laterally compressed, with 2–7 florets, generally with fertile florets (bisexual) only; the lowest floret is bisexual	Solitary, laterally compressed, with (2–)3–10 florets, with sterile florets (neuter or incomplete) both distal and proximal to the fertile florets; lowest floret male or sterile	Clusters, strongly laterally compressed, with 2–6 florets; bisexual, or bisexual and sterile (reduced, sterile spikelets are often present at the bases of the spikelet clusters)
Glumes	Subequal, as long as spikelets and lowest floret, 3 or 5 veined	Unequal, shorter than spikelets and lowest floret, 3–5 veined	Subequal, generally shorter than the spikelet and the lowest floret, 1 veined
Rachilla	Glabrous	With long white hairs	Glabrous
Lemmas	With long white hairs (silky hairs) on the back (outside) in their lower part, not keeled, 3–9 veined	Glabrous, not keeled, 1–3 veined	Glabrous or scabrid, ciliate on the keel and margins, keeled, 3 veined
Floret callus	Short (not elongated), with hairs	Linear (elongated), with long white hairs	Absent

**Table 2 plants-12-01850-t002:** Measurements of the vegetative and reproductive parts of *Arundo donax*.

Measurements	Units	Ranges	Mean ± SD
Culm height	m	3.0–8.2	6.10 ± 1.15
Number of culm internodes per culm	number	(35–)55–125	74.83 ± 23.92
Culm internode length	cm	1.7–17.0	7.30 ± 3.75
Basal culm internode length	cm	2.5–17.0	11.67 ± 2.89
Middle culm internode length	cm	4.7–15.7	9.13 ± 2.59
Apical culm internode length	cm	1.7–10.5	4.59 ± 1.81
Basal culm internode diameter	cm	1.54–3.28 (=1.5–3.3)	2.25 ± 0.47
Middle culm internode diameter	cm	1.25–2.89 (=1.3–2.9)	1.85 ± 0.45
Apical culm internode diameter	cm	0.35–2.45 (=0.4–2.5)	1.21 ± 0.43
Basal culm internode wall thickness	mm	1.4–7.4	3.88 ± 1.36
Middle culm internode wall thickness	mm	0.9–3.8	1.87 ± 0.51
Apical culm internode wall thickness	mm	0.3–1.9	1.04 ± 0.35
Leaf blade length	cm	24.0–73.2	45.78 ± 8.36
Leaf blade width	cm	2.0–11.7	4.95 ± 2.18
Leaf blade length/width ratio	ratio	4.6–14.7	10.24 ± 2.07
Branch leaf blade length	cm	12.0–55.0	35.77 ± 13.47
Branch leaf blade width	cm	1.0–7.0	3.77 ± 1.77
Branch leaf blade length/width ratio	ratio	5.6–11.6	8.50 ± 1.15
Ligule length	mm	1.0–2.3	1.40 ± 0.29
Leaf sheath length	cm	7.0–13.5	10.44 ± 1.51
Inflorescence length (including peduncle)	cm	70.0–165.0	119.10 ± 23.94
Inflorescence length (excluding peduncle)	cm	42.0–80.0	63.74 ± 8.86
Inflorescence width	cm	17.0–54.0	35.84 ± 12.53
Peduncle length	cm	26.0–95.0	55.35 ± 22.22
Basal peduncle diameter	cm	0.73–2.06 (= 0.7–2.1)	1.36 ± 0.38
Middle peduncle diameter	cm	0.69–1.77 (= 0.7–1.8)	1.19 ± 0.27
Primary branches length of inflorescences	cm	(5.0–)10.0–43.0	24.91 ± 9.66
Spikelet length	mm	6.0–16.0	9.67 ± 3.17
Spikelet width	mm	1.5–6.0	3.34 ± 1.35
Pedicel length	mm	1.5–10.0	4.38 ± 2.10
Glume length	mm	5.0–13.5	8.11 ± 2.00
Glume width	mm	0.6–1.6	1.10 ± 0.22
Lower glume length	mm	5.0–13.5	8.24 ± 2.17
Lower glume width	mm	0.6–1.6	1.10 ± 0.24
Upper glume length	mm	5.0–12.0	7.99 ± 1.84
Upper glume width	mm	0.6–1.6	1.11 ± 0.19
Lemma length (including awn)	mm	4.0–14.5	8.22 ± 2.76
Lemma width	mm	1.0–1.8	1.35 ± 0.19
Palea length	mm	2.5–6.0	4.03 ± 0.96

**Table 3 plants-12-01850-t003:** Measurements of the vascular bundles, vessels, and parenchyma cells of the culm internodes of *Arundo donax*.

Measurements	Units	Ranges	Mean ± SD
Outer small vascular bundle length	µm	45.18–121.80	77.31 ± 17.16
Outer small vascular bundle width	µm	54.03–134.64	99.58 ± 19.54
Outer small vessel length	µm	23.92–48.90	35.85 ± 6.95
Outer small vessel width	µm	21.48–42.83	31.34 ± 5.49
Larger vascular bundle length in ground tissue (including bundle sheath ring)	µm	146.98–285.45	225.84 ± 31.18
Larger vascular bundle width in ground tissue (including bundle sheath ring)	µm	168.87–355.25	262.77 ± 37.70
Larger vessel length	µm	64.81–113.27	95.86 ± 12.69
Larger vessel width	µm	44.97–103.40	82.33 ± 13.55
Density of the vascular bundles in ground tissue	number/mm^2^	7–14	10.27 ± 1.65
Parenchyma cell length in ground tissue	µm	37.59–93.97	63.73 ± 12.95
Parenchyma cell width in ground tissue	µm	37.31–75.84	56.12 ± 8.79

## Data Availability

All relevant data can be found within the manuscript.

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
