# Peer review of "Morphology, Taxonomy, Culm Internode and Leaf Anatomy, and Palynology of the Giant Reed (Arundo donax L.), Poaceae, Growing in Thailand"

_plants, 2023, doi:10.3390/plants12091850_

Round 1

Reviewer 1 Report

The objective is too general, and the manuscript is entirely descriptive. The description of the species is adequate and complete. The anatomy plates are well described, except for figure 12.

Too many figures are included, and at least figures 4, 9, and 10 do not add to the manuscript. 

The discussion is quite limited. The conclusions do not correspond with the results.

Author Response

Respond to Reviewer 1

For improvements and corrections in this paper, the authors highlighted in red.

The authors are grateful to thank the reviewer 1 for valuable comments.

Open Review

(x) I would like to sign my review report

Thank you very much for your kind consideration.

Quality of English Language

(x) Moderate English changes required

Dr Tushar Andriyas has checked English language for this paper.

Yes      Can be improved        Must be improved      Not applicable

Does the introduction provide sufficient background and include all relevant references?

( )        (x)       ( )        ( )

The authors added more information about the economic uses of the Arundo donax in the second paragraph of the introduction part.

The culms of Arundo donax are a useful source of cane for light construction, and for making woodwind instrument reeds [6,7]. Clarinet and other woodwind instruments use a thin strip of cane (culm internode), called a reed, to produce their sound. The reed is placed within the instrument’s mouthpiece where it vibrates according to the blowing pressure generated by the player. The anatomical characteristics of the culm internodes affect the musical performance of clarinet reeds made from A. donax [8]. Reeds of clarinet, oboe, and bassoon are mainly produced from the giant reed, A. donax. The best reed canes grow only in few areas of the Var in France due to its very mild Mediterranean climate, with lots of sunshine and the mistral [9].

Are all the cited references relevant to the research?

( )        ( )        (x)       ( )

The authors confirm that all the cited references relevant to the research.

Is the research design appropriate?

( )        ( )        ( )        (x)

The authors confirm that the research design is appropriate in agreement with reviewer 3.

Are the methods adequately described?

( )        ( )        (x)       ( )

The authors confirm that the methods are adequately described.

Are the results clearly presented?

(x)       ( )        ( )        ( )

Are the conclusions supported by the results?

( )        ( )        (x)       ( )

The authors added information in the conclusions.

Comments and Suggestions for Authors

The objective is too general, and the manuscript is entirely descriptive. The description of the species is adequate and complete.

This paper was conducted from a basic research subproject entitled “Morphology, Taxonomy, Anatomy, Palynology, Distribution, and Ecology of Arundo. donax L. (Poaceae) in                  Thailand” under the research project entitled “Study and Quality Development of Reed in Thailand for Clarinet Reed Production and their Utilization in Music Industry”.

The anatomy plates are well described, except for figure 12.

Thank you very much.

Figure 12. Transverse sections of leaf blades of Arundo donax. (A) whole leaf blade; (B) median and lateral vascular bundles; (C, D) second-order vascular bundles; (E) third-order vascular bundle. Abe: abaxial epidermis, Ade: adaxial epidermis, Bc: bulliform cells, Ibs: inner bundle sheath, Me: mesophyll, Obs: outer bundle sheath, Scg: sclerenchyma girder. Photos: Pichet Chanton and Thirawat Thaepthup.

The authors think full detailed description is shown in the result.

Too many figures are included, and at least figures 4, 9, and 10 do not add to the manuscript. 

May the authors add Figures 4, 9, and 10 to the manuscript.

In Figure 4, the authors would like to show characters of vegetative parts.

In Figure 9, the authors would like to show many habitats of Arundo donax in Thailand.

In Figure 10, the authors would like to show uses of Arundo donax in Thailand.

The discussion is quite limited. The conclusions do not correspond with the results.

The authors revised the discussion.

Discussion about culm internode anatomy and leaf blade anatomy is mentioned under the result.

The authors added information in the conclusions.

The morphology, taxonomy, culm internode and leaf blade anatomy, and palynology of Arundo donax are reported. Arundo is related to Phragmites, but differs in having its lemma with long white hairs outside below the middle part (vs glabrous); rachilla glabrous (vs with long white hairs); glumes subequal, as long as spikelet (vs unequal, shorter than spikelet); and lowest floret bisexual (vs male or sterile). Two names in Arundo are lectotypified: A. bifaria and A. bengalensis, and are synonyms of A. donax. It is mostly found in the wet habitats such as open areas along the river banks, streams, up to elevations of 620 m above mean sea level, especially in the northern region of Thailand. The culms are locally used for light construction, building rough fences, walls and benches, and are also used in the traditional Lanna rituals of “Suep Chata”. A. donax is also cultivated as a medicinal grass, and is used in the treatment of itching rash. It is also cultivated in the garden of Suan Luang Rama IX as an ornamental grass, to provide botanical education for people. Dr Nattapon Banjatammanon, a lecturer and clarinet player from the Department of Music, Faculty of Humanities, Kasetsart University, and a co-author in this paper, has been purchasing dried culms of A. donax from France for making clarinet reeds himself. The anatomical characteristics of the culm internodes affect the musical performance of clarinet reeds made from A. donax. The chloroplasts in the transverse section of the leaf blades are not found in the bundle sheath cells, which indicates that it is a C3 grass. Generally, the pollen grains in Poaceae are spheroidal with a single pore (monoporate) surrounded by an annulus and the exine sculpturing is granular.

Respond to Reviewers and Editor

For improvements and corrections, the authors highlighted in red.

Corrections

Line

Original

Corrections

24

habitat

habitat and ecology

30–31

The culm internodes have numerous…

The culm internodes in the                 transverse section have numerous

39–40

P/E ratio

polar axis length/equatorial axis length ratio (P/E ratio)

43

lectotypification

lectotype

48

of the subfamily Arundinoideae

under the subfamily Arundinoideae

71–72

-

added: The culms of Arundo donax are a useful source of cane for light construction, and for making woodwind instrument reeds [6,7].

82

In this paper we

In this paper, we

94–95

Amphidonax bengalensis (Retz.) Nees ex Steud.

Amphidonax bengalensis (Retz.) Steud.

95

W.D.J. Koch

W. D. J. Koch

Table 1.

Leaves / up to c. 8 cm wide

up to 11.7 cm wide

113–145

about author names

revised spacing for author names

206–207

A. donax

Arundo donax

Figure 4

(E, F)

(E)

Figure 4

(G)

(F)

268–269

Chaiyaphum [Thep Sathit (Ban Rai Subdistr.)]; CENTRAL:

Chaiyaphum [Thep Sathit (Ban Rai Subdistr.)], Nakhon Ratchasima [Pak Chong (Nong Sarai)]; SOUTH-WESTERN: Kanchanaburi [Thong Pha Phum (Linthin Subdistr.)];            CENTRAL:

Figure 8

The map has changed because a location is added.

311, 334–343

about author names

revised spacing for author names

346

Ngernsaengsaruay et al. Ad11-09022023

C. Ngernsaengsaruay et al. Ad11-09022023

348

Ngernsaengsaruay et al. Ad12-09022023

C. Ngernsaengsaruay et al. Ad12-09022023

357–362

K. Kertsawang 3241 (BK, QBG)]; CENTRAL:

K. Kertsawang 3241 (BK, QBG)]; Nakhon Ratchasima [Nong Sarai Subdistr., Pak Chong Distr., 14° 38' 39.84'' N, 101° 30' 43.83'' E, 350 m alt., sterile specimen, 10 March 2023, C. Ngernsaengsaruay & P. Chanton Ad13-10032023 (BK, BKF, QBG)]; SOUTH-WESTERN: Kanchanaburi [Mae Klong Watershed Research Station, Linthin Subdistr., Thong Pha Phum Distr., in open area along stream (Professor Dr Dokrak Marod own observation; CENTRAL:

370

Ngernsaengsaruay et al. Ad01-15122021

C. Ngernsaengsaruay et al. Ad01-15122021

371

Bangkok, cultivated,

Bangkok (locality not specified), cultivated,

371–452

about author names

revised spacing for author names

458

Panjab

Punjab

466

A. donax

Arundo donax

Figure 9

Ban Ai, Chai Prakan Subdistrict, Chai Prakan District

Ban Ai, Si Dong Yen Subdistrict, Chai Prakan District

494

A. donax

Arundo donax

501

Mae Hong Son and Chiang Mai Provinces

Mae Hong Son, Chiang Mai and Lamphun Provinces

516

The previous studies

Previous studies

516

A. donax

Arundo donax

534

A. donax

Arundo donax

Figure 10

(C) used in building walls and benches;

(C) used in building rough walls and benches;

545

A. donax

Arundo donax

568

A. donax

Arundo donax

Figure 12

 (C, D)

12. (C, D)

Figure 13

information added: Cr: styloid crystals

602

A. donax

Arundo donax

604

A. donax

Arundo donax

609

information added: The styloid crystals can be found in the leaf blade epidermal cells.

619

A. donax

Arundo donax

627

A. donax

Arundo donax

634–636

Backer & Bakhuizen van den Brink [11] and Watson & Dallwitz [20] reported the branches of the genus Arundo are extravaginal, but from our study, we found the                    intravaginal branching in A. donax (Figure 4B and 4C).

According to Backer & Bakhuizen van den Brink [13] and Watson & Dallwitz [7], the branches of the genus Arundo are extravaginal; however, from our observations, we found the branches of Arundo donax are intravaginal (Figure 4B and 4C).

641–644

The previous studies reported the size of leaf blades of this species: c. 40 × 1.5–4 cm [18], 30–60 × 2–5 cm [19], 30–60 × 2.5–5 cm [15], 15–75 × 0.8–4.5 cm [11], up to 75 × 4 cm [12], 50–70(–90) × 2–6 cm [17] and up to 1 m × 6 cm [32].

According to previous studies, the sizes of leaf blades are c. 40 × 1.5–4 cm [19], 30–60 × 2–5 cm [20], 30–60 × 2.5–5 cm [17], 15–75 × 0.8–4.5 cm [13], up to 75 × 4 cm [14], 50–70(–90) × 2–6 cm [18] and up to 1 m × 6 cm [32]. Furthermore, from our study, we found the size of leaf blades ranges from 24–73.2 × 2–11.7 cm.,

644

sometimes wider than the previous studies.

sometimes wider than previous studies.

646–651

The previous studies reported the length of inflorescences of this species: 30–50 cm [10], 30–50 cm long or more [12], 30–60 cm [8,15,19], up to 60 cm [32], 30–70 cm [17] and 30–75 cm [11].

According to previous studies, the lengths of inflorescences are 30–50 cm [12], 30–50 cm long or more [14], 30–60 cm [10,17,20], up to 60 cm [32], 30–70 cm [18] and 30–75 cm [13]. In addition, we found the length of inflorescences ranges from 70–165 cm (including peduncle), 42–80 cm (excluding peduncle) in this study, sometimes longer than previous studies. The lengths of primary branches of inflorescences are 10–25 cm [20] and 20–30 cm (longest) [19], but we found (5–)10–43 cm long in this study,              sometimes longer than previous studies.

646–652

The lengths of primary branches of inflorescences of this species are 10–25 cm [19] and 20–30 cm (longest) [18], but we found (5–)10–43 cm long in this study, sometimes longer than the previous studies.

The lengths of primary branches of inflorescences are 10–25 cm [19] and 20–30 cm (longest) [18], but we found (5–)10–43 cm long in this study, sometimes longer than previous studies.

664–666

The previous studies reported the size of pollens of this species: 12 × 12 µm [33] and 29–40 µm in diam. [34].

According to previous studies, the sizes of pollens are 12 × 12 µm [33] and 29–40 µm in diam. [34]. Furthermore, from our study, we found the pollen diameter in equatorial axis ranges from 24.95–38.13 (29.52 ± 3.79) µm in consistent with Trigo & Fernández [34].

667–668

Sanghi & Sarna [33] reported the exine sculpturing of this species is psilate, but from our study, we found the granular.

According to Sanghi & Sarna [33], the exine sculpturing is psilate; however, from our observations, we found the granular.

671

A. donax

Arundo donax

672

Plant specimens of A. donax were observed and collected in the northern (Mae Hong Son, Chiang Mai, Lamphun and Nakhon Sawan) and the central……

Plant specimens of A. donax were observed and collected in the northern (Mae Hong Son, Chiang Mai, Lamphun and Nakhon Sawan), the eastern (Nakhon Ratchasima) and the central……

676–677

information added: The herbarium accession number can be seen in the specimens examined.

695

information added: The anatomy terminologies follow Metcalfe [39].

697

(Ngernsaengsaruay et al. Ad01-15122021)

(C. Ngernsaengsaruay et al. Ad01-15122021)

702–704

information added: The characteristics of pollen grains (size, shape, symmetry, aperture, and exine thickness and sculpturing) are examined and measured. The pollen morphology terminologies follow Punt et al. [40].

705–726

information added: see Conclusions

727

A. donax

Arundo donax

749

A. donax

Arundo donax

750–751

A. donax deposited in QBG.

A. donax deposited in QBG and Professor Dr Dokrak Marod for his own observation of A. donax in Kanchanaburi Province.

754

in the northern and the central regions

in the northern, the eastern and the central regions

Table A1.

p. 31

Ban Ai, Chai Prakan Subdistr., Chai Prakan Distr.,

Ban Ai, Si Dong Yen Subdistr., Chai Prakan Distr.,

Table A1.

p. 31

-

Information is added

References

The order of references from 6–20 was changed.

References

Reference no 39 and 40 are added.

Reviewer 2 Report

This manuscript describes the Morphology, Taxonomy, Culm Internode and Leaf Anatomy, and Palynology of the Giant Reed (Arundo donax L.), belong to Poaceae family, growing in Thailand (Poaceae). Culm anatomical studies are highly welcomed to help identify and delimit grass taxa. Although the manuscript's premise and goal are reliable, the interpretation of the results and the study's greater context are lacking in many ways.

Since I do find some merit in the manuscript, I would be willing to reconsider if you wish to undertake major revisions and re-submit, addressing these concerns.

The problem of the study is unclear. As mentioned above, the morphology has already been described before. The study should therefore focus mainly on micromorphological characters never published before. Clarifying phylogenetic relationships requires the reconstruction of a phylogeny, which is not done here.

Botanical names should not be italicized in some places at the manuscript.

The aim of the paper is still vague as it is reflected in the poorly written.

The conclusion section of the abstract should be restructured to describe the implication of the studied characters. In the introduction, the authors did not adequately support their declarations about the correlation between the Arundo genus species.

Mentioned the herbarium specimens’ accession number in the methodology section to make the study authentic.

Pollen morphometry was described using scanning electron microscopy but the methods of the study were described. Describe the authenticated procedure used in the laboratory.  

Still, the poor language with convoluted often barely understandable sentences. First of all the English form of the text is not acceptable, sentences are hard to be understood and in some cases, uncorrected sentences are also present. The English version of the paper should be strongly revised by a mother tongue Professor, to facilitate the comprehension of the concepts. The manuscript language is awkward and would require thorough proofreading by a proficient English speaker.

The major omission of this manuscript is not to follow the standard terminology of Poaceae. Descriptions of gross anatomy must follow Metcalfe (1960), Goller (1977), Clark and Fisher (1987), and Kellogg (2015). In this sense, the authors must review the terminology employed throughout the manuscript. Addressing the standard terminology is mandatory in grass anatomy studies to ensure data reusability.

The central premise of this work was to seek a Morpho-taxonomic approach to visualize macro and micromorphological characters to assist in the taxonomy of grass species. However, even though the authors highlight differences among the studied species in Table 1, their interpretation of the results is severely lacking. The discussion needs to be centered on understanding the importance of micromorphological features to accurately identify the grass species. Previous works have documented many of the reported observations. In this sense, several paragraphs in the Discussion section only bring comparisons and are inconclusive or speculative. How does this study contribute to new knowledge? What should matter the most to the readers after they have finished reading the paper? What culm anatomical features do the authors recommend for taxonomic purposes?

Finally, it is unclear how the authors concluded (based on its economic and marketing value) that the observed morpho-anatomical and palynological features can be associated with traditional medicines. This should not be the focus of the manuscript. Further arguments and evidence for differences related to "fitness" need to come from more specific studies on statistical analysis via UPGMA dendrogram and PCA clustering analysis to correlate the characters of the grass species.

Author Response

Respond to Reviewer 2

For improvements and corrections in this paper, the authors highlighted in red.

The authors are grateful to thank the reviewer 2 for valuable comments.

Thank you very much for your kind consideration.

Open Review

(x) I would not like to sign my review report

Quality of English Language

Reviewer 2

(x) Extensive editing of English language and style required

Yes      Can be improved        Must be improved      Not applicable

Does the introduction provide sufficient background and include all relevant references?

( )        (x)       ( )        ( )

The authors added more information about the economic uses of the Arundo donax in the second paragraph of the introduction part.

The culms of Arundo donax are a useful source of cane for light construction, and for making woodwind instrument reeds [6,7].

Are all the cited references relevant to the research?

( )        (x)       ( )        ( )

The authors confirm that all the cited references relevant to the research.

Is the research design appropriate?

(x)       ( )        ( )        ( )

Are the methods adequately described?

( )        (x)       ( )        ( )

The authors added information in the methods.

Are the results clearly presented?

(x)       ( )        ( )        ( )

Are the conclusions supported by the results?

( )        ( )        (x)       ( )

The authors added information in the conclusions.

Comments and Suggestions for Authors

This manuscript describes the Morphology, Taxonomy, Culm Internode and Leaf Anatomy, and Palynology of the Giant Reed (Arundo donax L.), belong to Poaceae family, growing in Thailand (Poaceae). Culm anatomical studies are highly welcomed to help identify and delimit grass taxa. Although the manuscript's premise and goal are reliable, the interpretation of the results and the study's greater context are lacking in many ways.

Since I do find some merit in the manuscript, I would be willing to reconsider if you wish to undertake major revisions and re-submit, addressing these concerns.

            The manuscript was revised by the authors.

This study provided the morphology, taxonomy, anatomy, and palynology of Arundo donax, for the first time. The authors presented all information such as a detailed morphological description, identification, distribution, the specimens examined, habitat, conservation assessment, phenology, etymology, vernacular name, and uses of the studied species. The results were well supported by tables, figures, and supplemental materials. The authors think the information in this paper is adequate to publish in the “Plants”.

The problem of the study is unclear. As mentioned above, the morphology has already been described before. The study should therefore focus mainly on micromorphological characters never published before. Clarifying phylogenetic relationships requires the reconstruction of a phylogeny, which is not done here.

Although the morphology has already been described before, but from our observations, The authors added more and more full detailed description and other information than others previous studies (please see the manuscript). The authors are supposing that the information in this paper is adequate to publish in the “Plants”.

Botanical names should not be italicized in some places at the manuscript.

The authors confirm that the botanical names are always italicized.

The aim of the paper is still vague as it is reflected in the poorly written.

The aim of this paper is in the last paragraph of the introduction.

The conclusion section of the abstract should be restructured to describe the implication of the studied characters. In the introduction, the authors did not adequately support their declarations about the correlation between the Arundo genus species.

The abstract was revised by the authors.

Mentioned the herbarium specimens’ accession number in the methodology section to make the study authentic.

            The authors mentioned the herbarium specimens’ accession number in Table A1 and additional specimens examined.

Plant specimens of Arundo donax were observed and collected in the northern (Mae Hong Son, Chiang Mai, Lamphun and Nakhon Sawan), the eastern (Nakhon Ratchasima) and the central (Chai Nat and Bangkok) regions of Thailand (Table A1, Figure A1). Herbarium specimens deposited in BK, BKF, QBG, and those included in the digital herbarium databases of BM, C, E, JSTOR, K, K-W, L and P were examined by consulting the taxonomic literature (acronyms follow Thiers [35]). The herbarium accession number can be seen in the specimens examined.

Pollen morphometry was described using scanning electron microscopy but the methods of the study were described. Describe the authenticated procedure used in the laboratory. 

The authors added this sentence “The characteristics of pollen grains (size, shape, symmetry, aperture, and exine thickness and sculpturing) are examined and measured. The pollen morphology terminologies follow Punt et al. [40]..” in the methods.

Still, the poor language with convoluted often barely understandable sentences. First of all the English form of the text is not acceptable, sentences are hard to be understood and in some cases, uncorrected sentences are also present. The English version of the paper should be strongly revised by a mother tongue Professor, to facilitate the comprehension of the concepts. The manuscript language is awkward and would require thorough proofreading by a proficient English speaker.

Dr Tushar Andriyas has checked English language for this paper.

English language in this paper following the plant taxonomy style and format. The authors agree with reviewer 3 and 1.

Reviewer 1

(x) Moderate English changes required

Reviewer 3

(x) English language and style are fine/minor spell check required

The major omission of this manuscript is not to follow the standard terminology of Poaceae. Descriptions of gross anatomy must follow Metcalfe (1960), Goller (1977), Clark and Fisher (1987), and Kellogg (2015). In this sense, the authors must review the terminology employed throughout the manuscript. Addressing the standard terminology is mandatory in grass anatomy studies to ensure data reusability.

The authors added this sentence “The anatomy terminologies follow Metcalfe [39].” in the methods.

The central premise of this work was to seek a Morpho-taxonomic approach to visualize macro and micromorphological characters to assist in the taxonomy of grass species. However, even though the authors highlight differences among the studied species in Table 1, their interpretation of the results is severely lacking. The discussion needs to be centered on understanding the importance of micromorphological features to accurately identify the grass species. Previous works have documented many of the reported observations. In this sense, several paragraphs in the Discussion section only bring comparisons and are inconclusive or speculative. How does this study contribute to new knowledge? What should matter the most to the readers after they have finished reading the paper? What culm anatomical features do the authors recommend for taxonomic purposes?

The manuscript was revised by the authors.

This study provided the morphology, taxonomy, anatomy, and palynology of Arundo donax, for the first time. The authors presented all information such as a detailed morphological description, identification, distribution, the specimens examined, habitat, conservation assessment, phenology, etymology, vernacular name, and uses of the studied species. The results were well supported by tables, figures, and supplemental materials. The authors think the information in this paper is adequate to publish in the “Plants”.

Finally, it is unclear how the authors concluded (based on its economic and marketing value) that the observed morpho-anatomical and palynological features can be associated with traditional medicines. This should not be the focus of the manuscript. Further arguments and evidence for differences related to "fitness" need to come from more specific studies on statistical analysis via UPGMA dendrogram and PCA clustering analysis to correlate the characters of the grass species.

The authors added information in the conclusions.

The morphology, taxonomy, culm internode and leaf blade anatomy, and palynology of Arundo donax are reported. Arundo is related to Phragmites, but differs in having its lemma with long white hairs outside below the middle part (vs glabrous); rachilla glabrous (vs with long white hairs); glumes subequal, as long as spikelet (vs unequal, shorter than spikelet); and lowest floret bisexual (vs male or sterile). Two names in Arundo are lectotypified: A. bifaria and A. bengalensis, and are synonyms of A. donax. It is mostly found in the wet habitats such as open areas along the river banks, streams, up to elevations of 620 m above mean sea level, especially in the northern region of Thailand. The culms are locally used for light construction, building rough fences, walls and benches, and are also used in the traditional Lanna rituals of “Suep Chata”. A. donax is also cultivated as a medicinal grass, and is used in the treatment of itching rash. It is also cultivated in the garden of Suan Luang Rama IX as an ornamental grass, to provide botanical education for people. Dr Nattapon Banjatammanon, a lecturer and clarinet player from the Department of Music, Faculty of Humanities, Kasetsart University, and a co-author in this paper, has been purchasing dried culms of A. donax from France for making clarinet reeds himself. The anatomical characteristics of the culm internodes affect the musical performance of clarinet reeds made from A. donax. The chloroplasts in the transverse section of the leaf blades are not found in the bundle sheath cells, which indicates that it is a C3 grass. Generally, the pollen grains in Poaceae are spheroidal with a single pore (monoporate) surrounded by an annulus and the exine sculpturing is granular.

Date of this review

20 Mar 2023

Respond to Reviewers and Editor

For improvements and corrections, the authors highlighted in red.

Corrections

Line

Original

Corrections

24

habitat

habitat and ecology

30–31

The culm internodes have numerous…

The culm internodes in the                 transverse section have numerous

39–40

P/E ratio

polar axis length/equatorial axis length ratio (P/E ratio)

43

lectotypification

lectotype

48

of the subfamily Arundinoideae

under the subfamily Arundinoideae

71–72

-

added: The culms of Arundo donax are a useful source of cane for light construction, and for making woodwind instrument reeds [6,7].

82

In this paper we

In this paper, we

94–95

Amphidonax bengalensis (Retz.) Nees ex Steud.

Amphidonax bengalensis (Retz.) Steud.

95

W.D.J. Koch

W. D. J. Koch

Table 1.

Leaves / up to c. 8 cm wide

up to 11.7 cm wide

113–145

about author names

revised spacing for author names

206–207

A. donax

Arundo donax

Figure 4

(E, F)

(E)

Figure 4

(G)

(F)

268–269

Chaiyaphum [Thep Sathit (Ban Rai Subdistr.)]; CENTRAL:

Chaiyaphum [Thep Sathit (Ban Rai Subdistr.)], Nakhon Ratchasima [Pak Chong (Nong Sarai)]; SOUTH-WESTERN: Kanchanaburi [Thong Pha Phum (Linthin Subdistr.)];            CENTRAL:

Figure 8

The map has changed because a location is added.

311, 334–343

about author names

revised spacing for author names

346

Ngernsaengsaruay et al. Ad11-09022023

C. Ngernsaengsaruay et al. Ad11-09022023

348

Ngernsaengsaruay et al. Ad12-09022023

C. Ngernsaengsaruay et al. Ad12-09022023

357–362

K. Kertsawang 3241 (BK, QBG)]; CENTRAL:

K. Kertsawang 3241 (BK, QBG)]; Nakhon Ratchasima [Nong Sarai Subdistr., Pak Chong Distr., 14° 38' 39.84'' N, 101° 30' 43.83'' E, 350 m alt., sterile specimen, 10 March 2023, C. Ngernsaengsaruay & P. Chanton Ad13-10032023 (BK, BKF, QBG)]; SOUTH-WESTERN: Kanchanaburi [Mae Klong Watershed Research Station, Linthin Subdistr., Thong Pha Phum Distr., in open area along stream (Professor Dr Dokrak Marod own observation; CENTRAL:

370

Ngernsaengsaruay et al. Ad01-15122021

C. Ngernsaengsaruay et al. Ad01-15122021

371

Bangkok, cultivated,

Bangkok (locality not specified), cultivated,

371–452

about author names

revised spacing for author names

458

Panjab

Punjab

466

A. donax

Arundo donax

Figure 9

Ban Ai, Chai Prakan Subdistrict, Chai Prakan District

Ban Ai, Si Dong Yen Subdistrict, Chai Prakan District

494

A. donax

Arundo donax

501

Mae Hong Son and Chiang Mai Provinces

Mae Hong Son, Chiang Mai and Lamphun Provinces

516

The previous studies

Previous studies

516

A. donax

Arundo donax

534

A. donax

Arundo donax

Figure 10

(C) used in building walls and benches;

(C) used in building rough walls and benches;

545

A. donax

Arundo donax

568

A. donax

Arundo donax

Figure 12

 (C, D)

12. (C, D)

Figure 13

information added: Cr: styloid crystals

602

A. donax

Arundo donax

604

A. donax

Arundo donax

609

information added: The styloid crystals can be found in the leaf blade epidermal cells.

619

A. donax

Arundo donax

627

A. donax

Arundo donax

634–636

Backer & Bakhuizen van den Brink [11] and Watson & Dallwitz [20] reported the branches of the genus Arundo are extravaginal, but from our study, we found the                    intravaginal branching in A. donax (Figure 4B and 4C).

According to Backer & Bakhuizen van den Brink [13] and Watson & Dallwitz [7], the branches of the genus Arundo are extravaginal; however, from our observations, we found the branches of Arundo donax are intravaginal (Figure 4B and 4C).

641–644

The previous studies reported the size of leaf blades of this species: c. 40 × 1.5–4 cm [18], 30–60 × 2–5 cm [19], 30–60 × 2.5–5 cm [15], 15–75 × 0.8–4.5 cm [11], up to 75 × 4 cm [12], 50–70(–90) × 2–6 cm [17] and up to 1 m × 6 cm [32].

According to previous studies, the sizes of leaf blades are c. 40 × 1.5–4 cm [19], 30–60 × 2–5 cm [20], 30–60 × 2.5–5 cm [17], 15–75 × 0.8–4.5 cm [13], up to 75 × 4 cm [14], 50–70(–90) × 2–6 cm [18] and up to 1 m × 6 cm [32]. Furthermore, from our study, we found the size of leaf blades ranges from 24–73.2 × 2–11.7 cm.,

644

sometimes wider than the previous studies.

sometimes wider than previous studies.

646–651

The previous studies reported the length of inflorescences of this species: 30–50 cm [10], 30–50 cm long or more [12], 30–60 cm [8,15,19], up to 60 cm [32], 30–70 cm [17] and 30–75 cm [11].

According to previous studies, the lengths of inflorescences are 30–50 cm [12], 30–50 cm long or more [14], 30–60 cm [10,17,20], up to 60 cm [32], 30–70 cm [18] and 30–75 cm [13]. In addition, we found the length of inflorescences ranges from 70–165 cm (including peduncle), 42–80 cm (excluding peduncle) in this study, sometimes longer than previous studies. The lengths of primary branches of inflorescences are 10–25 cm [20] and 20–30 cm (longest) [19], but we found (5–)10–43 cm long in this study,              sometimes longer than previous studies.

646–652

The lengths of primary branches of inflorescences of this species are 10–25 cm [19] and 20–30 cm (longest) [18], but we found (5–)10–43 cm long in this study, sometimes longer than the previous studies.

The lengths of primary branches of inflorescences are 10–25 cm [19] and 20–30 cm (longest) [18], but we found (5–)10–43 cm long in this study, sometimes longer than previous studies.

664–666

The previous studies reported the size of pollens of this species: 12 × 12 µm [33] and 29–40 µm in diam. [34].

According to previous studies, the sizes of pollens are 12 × 12 µm [33] and 29–40 µm in diam. [34]. Furthermore, from our study, we found the pollen diameter in equatorial axis ranges from 24.95–38.13 (29.52 ± 3.79) µm in consistent with Trigo & Fernández [34].

667–668

Sanghi & Sarna [33] reported the exine sculpturing of this species is psilate, but from our study, we found the granular.

According to Sanghi & Sarna [33], the exine sculpturing is psilate; however, from our observations, we found the granular.

671

A. donax

Arundo donax

672

Plant specimens of A. donax were observed and collected in the northern (Mae Hong Son, Chiang Mai, Lamphun and Nakhon Sawan) and the central……

Plant specimens of A. donax were observed and collected in the northern (Mae Hong Son, Chiang Mai, Lamphun and Nakhon Sawan), the eastern (Nakhon Ratchasima) and the central……

676–677

information added: The herbarium accession number can be seen in the specimens examined.

695

information added: The anatomy terminologies follow Metcalfe [39].

697

(Ngernsaengsaruay et al. Ad01-15122021)

(C. Ngernsaengsaruay et al. Ad01-15122021)

702–704

information added: The characteristics of pollen grains (size, shape, symmetry, aperture, and exine thickness and sculpturing) are examined and measured. The pollen morphology terminologies follow Punt et al. [40].

705–726

information added: see Conclusions

727

A. donax

Arundo donax

749

A. donax

Arundo donax

750–751

A. donax deposited in QBG.

A. donax deposited in QBG and Professor Dr Dokrak Marod for his own observation of A. donax in Kanchanaburi Province.

754

in the northern and the central regions

in the northern, the eastern and the central regions

Table A1.

p. 31

Ban Ai, Chai Prakan Subdistr., Chai Prakan Distr.,

Ban Ai, Si Dong Yen Subdistr., Chai Prakan Distr.,

Table A1.

p. 31

-

Information is added

References

The order of references from 6–20 was changed.

References

Reference no 39 and 40 are added.

Reviewer 3 Report

Manuscript ID: plants-2285072

Morphology, Taxonomy, Culm Internode and Leaf Anatomy, and Palynology of the Giant Reed (Arundo donax L.), Poaceae, Growing in Thailand

By Chatchai Ngernsaengsaruay et al.

Overview: This study provided the morphology, taxonomy, anatomy, and palynology of Arundo donax, for the first time. The authors presented all information such as a detailed morphological description, identification, distribution, the specimens examined, habitat, conservation assessment, phenology, etymology, vernacular name, and uses of the studied species. English writing is good, and results were well supported by tables, figures, and supplemental materials. In general, this manuscript is quite exciting, and I would like to see the paper published in the Plants’. I think minor revisions should be needed for this paper to publish.

Point 1: Please provide the economic uses of the Arundo donax in the introduction part.

Point 2: Please check the authority of the scientific name of the homotypic and heterotypic synonyms following the standard abbreviation (IPNI Life Sciences Identifier (LSID))

Point 3: Please provide the micro-images using SEM of the Arundo donax leaf if possible.

Point 4: The phytoliths or crystals are the most important characters for identification in Poaceae (Grasses). Moreover, the author can find crystals in Figure 13. Please provide the feature of the phytoliths or crystals.

Author Response

Respond to Reviewer 3

For improvements and corrections in this paper, the authors highlighted in red.

The authors are grateful to thank the reviewer 3 for valuable comments.

Thank you very much for your kind consideration.

Open Review

(x) I would not like to sign my review report

Quality of English Language

(x) English language and style are fine/minor spell check required

Dr Tushar Andriyas has checked English language for this paper.

Yes      Can be improved        Must be improved      Not applicable

Does the introduction provide sufficient background and include all relevant references?

( )        (x)       ( )        ( )

The authors added information as your comment.

Are all the cited references relevant to the research?

( )        (x)       ( )        ( )

The authors confirm that all the cited references relevant to the research.

Is the research design appropriate?

(x)       ( )        ( )        ( )

Are the methods adequately described?

(x)       ( )        ( )        ( )

Are the results clearly presented?

(x)       ( )        ( )        ( )

Are the conclusions supported by the results?

( )        (x)       ( )        ( )

The authors added information in the conclusions.

Comments and Suggestions for Authors

Morphology, Taxonomy, Culm Internode and Leaf Anatomy, and Palynology of the Giant Reed (Arundo donax L.), Poaceae, Growing in Thailand

By Chatchai Ngernsaengsaruay et al.

Overview: This study provided the morphology, taxonomy, anatomy, and palynology of Arundo donax, for the first time. The authors presented all information such as a detailed morphological description, identification, distribution, the specimens examined, habitat, conservation assessment, phenology, etymology, vernacular name, and uses of the studied species. English writing is good, and results were well supported by tables, figures, and supplemental materials. In general, this manuscript is quite exciting, and I would like to see the paper published in the ‘Plants’. I think minor revisions should be needed for this paper to publish.

Point 1: Please provide the economic uses of the Arundo donax in the introduction part.

The authors have mentioned the economic uses of the Arundo donax in the second paragraph of the introduction part, and added more information in the first line.

The culms of Arundo donax are a useful source of cane for light construction, and for making woodwind instrument reeds [6,7]. Clarinet and other woodwind instruments use a thin strip of cane (culm internode), called a reed, to produce their sound. The reed is placed within the instrument’s mouthpiece where it vibrates according to the blowing pressure generated by the player. The anatomical characteristics of the culm internodes affect the musical performance of clarinet reeds made from A. donax [8]. Reeds of clarinet, oboe, and bassoon are mainly produced from the giant reed, A. donax. The best reed canes grow only in few areas of the Var in France due to its very mild Mediterranean climate, with lots of sunshine and the mistral [9].

Point 2: Please check the authority of the scientific name of the homotypic and heterotypic synonyms following the standard abbreviation (IPNI Life Sciences Identifier (LSID))

The authors have checked the history of the scientific name “Amphidonax bengalensis (Retz.) Nees ex Steud.” from the publications, it is not Amphidonax bengalensis (Retz.) Steud. as IPNI mentioned. The authority of the scientific name should be “Amphidonax bengalensis (Retz.) Nees ex Steud.”

Point 3: Please provide the micro-images using SEM of the Arundo donax leaf if possible.

The authors think the information of leaf blade anatomy of Arundo donax are adequately described in this paper. We have not much time to do additional laboratory, this may cause the condition of funding source.

Point 4: The phytoliths or crystals are the most important characters for identification in Poaceae (Grasses). Moreover, the author can find crystals in Figure 13. Please provide the feature of the phytoliths or crystals.

The authors added the sentence “The styloid crystals can be found in the leaf blade epidermal cells.” in the second paragraph of leaf blade anatomy, and also added labelled in Figure 13.

Figure 13. Leaf blade epidermis of Arundo donax. (AD) abaxial surfaces; (E) adaxial surface. Cr: styloid crystals, Cz: costal zone, Gc: guard cells, Iz: intercostal zone, Lc: long cells, Shc: short cells, St: stoma, Suc: subsidiary cells. Photos: Pichet Chanton and Thirawat Thaepthup.

Conclusions

The morphology, taxonomy, culm internode and leaf blade anatomy, and palynology of Arundo donax are reported. Arundo is related to Phragmites, but differs in having its lemma with long white hairs outside below the middle part (vs glabrous); rachilla glabrous (vs with long white hairs); glumes subequal, as long as spikelet (vs unequal, shorter than spikelet); and lowest floret bisexual (vs male or sterile). Two names in Arundo are lectotypified: A. bifaria and A. bengalensis, and are synonyms of A. donax. It is mostly found in the wet habitats such as open areas along the river banks, streams, up to elevations of 620 m above mean sea level, especially in the northern region of Thailand. The culms are locally used for light construction, building rough fences, walls and benches, and are also used in the traditional Lanna rituals of “Suep Chata”. A. donax is also cultivated as a medicinal grass, and is used in the treatment of itching rash. It is also cultivated in the garden of Suan Luang Rama IX as an ornamental grass, to provide botanical education for people. Dr Nattapon Banjatammanon, a lecturer and clarinet player from the Department of Music, Faculty of Humanities, Kasetsart University, and a co-author in this paper, has been purchasing dried culms of A. donax from France for making clarinet reeds himself. The anatomical characteristics of the culm internodes affect the musical performance of clarinet reeds made from A. donax. The chloroplasts in the transverse section of the leaf blades are not found in the bundle sheath cells, which indicates that it is a C3 grass. Generally, the pollen grains in Poaceae are spheroidal with a single pore (monoporate) surrounded by an annulus and the exine sculpturing is granular.

Date of this review

07 Apr 2023

Respond to Reviewers and Editor

For improvements and corrections, the authors highlighted in red.

Corrections

Line

Original

Corrections

24

habitat

habitat and ecology

30–31

The culm internodes have numerous…

The culm internodes in the                 transverse section have numerous

39–40

P/E ratio

polar axis length/equatorial axis length ratio (P/E ratio)

43

lectotypification

lectotype

48

of the subfamily Arundinoideae

under the subfamily Arundinoideae

71–72

-

added: The culms of Arundo donax are a useful source of cane for light construction, and for making woodwind instrument reeds [6,7].

82

In this paper we

In this paper, we

94–95

Amphidonax bengalensis (Retz.) Nees ex Steud.

Amphidonax bengalensis (Retz.) Steud.

95

W.D.J. Koch

W. D. J. Koch

Table 1.

Leaves / up to c. 8 cm wide

up to 11.7 cm wide

113–145

about author names

revised spacing for author names

206–207

A. donax

Arundo donax

Figure 4

(E, F)

(E)

Figure 4

(G)

(F)

268–269

Chaiyaphum [Thep Sathit (Ban Rai Subdistr.)]; CENTRAL:

Chaiyaphum [Thep Sathit (Ban Rai Subdistr.)], Nakhon Ratchasima [Pak Chong (Nong Sarai)]; SOUTH-WESTERN: Kanchanaburi [Thong Pha Phum (Linthin Subdistr.)];            CENTRAL:

Figure 8

The map has changed because a location is added.

311, 334–343

about author names

revised spacing for author names

346

Ngernsaengsaruay et al. Ad11-09022023

C. Ngernsaengsaruay et al. Ad11-09022023

348

Ngernsaengsaruay et al. Ad12-09022023

C. Ngernsaengsaruay et al. Ad12-09022023

357–362

K. Kertsawang 3241 (BK, QBG)]; CENTRAL:

K. Kertsawang 3241 (BK, QBG)]; Nakhon Ratchasima [Nong Sarai Subdistr., Pak Chong Distr., 14° 38' 39.84'' N, 101° 30' 43.83'' E, 350 m alt., sterile specimen, 10 March 2023, C. Ngernsaengsaruay & P. Chanton Ad13-10032023 (BK, BKF, QBG)]; SOUTH-WESTERN: Kanchanaburi [Mae Klong Watershed Research Station, Linthin Subdistr., Thong Pha Phum Distr., in open area along stream (Professor Dr Dokrak Marod own observation; CENTRAL:

370

Ngernsaengsaruay et al. Ad01-15122021

C. Ngernsaengsaruay et al. Ad01-15122021

371

Bangkok, cultivated,

Bangkok (locality not specified), cultivated,

371–452

about author names

revised spacing for author names

458

Panjab

Punjab

466

A. donax

Arundo donax

Figure 9

Ban Ai, Chai Prakan Subdistrict, Chai Prakan District

Ban Ai, Si Dong Yen Subdistrict, Chai Prakan District

494

A. donax

Arundo donax

501

Mae Hong Son and Chiang Mai Provinces

Mae Hong Son, Chiang Mai and Lamphun Provinces

516

The previous studies

Previous studies

516

A. donax

Arundo donax

534

A. donax

Arundo donax

Figure 10

(C) used in building walls and benches;

(C) used in building rough walls and benches;

545

A. donax

Arundo donax

568

A. donax

Arundo donax

Figure 12

 (C, D)

12. (C, D)

Figure 13

information added: Cr: styloid crystals

602

A. donax

Arundo donax

604

A. donax

Arundo donax

609

information added: The styloid crystals can be found in the leaf blade epidermal cells.

619

A. donax

Arundo donax

627

A. donax

Arundo donax

634–636

Backer & Bakhuizen van den Brink [11] and Watson & Dallwitz [20] reported the branches of the genus Arundo are extravaginal, but from our study, we found the                    intravaginal branching in A. donax (Figure 4B and 4C).

According to Backer & Bakhuizen van den Brink [13] and Watson & Dallwitz [7], the branches of the genus Arundo are extravaginal; however, from our observations, we found the branches of Arundo donax are intravaginal (Figure 4B and 4C).

641–644

The previous studies reported the size of leaf blades of this species: c. 40 × 1.5–4 cm [18], 30–60 × 2–5 cm [19], 30–60 × 2.5–5 cm [15], 15–75 × 0.8–4.5 cm [11], up to 75 × 4 cm [12], 50–70(–90) × 2–6 cm [17] and up to 1 m × 6 cm [32].

According to previous studies, the sizes of leaf blades are c. 40 × 1.5–4 cm [19], 30–60 × 2–5 cm [20], 30–60 × 2.5–5 cm [17], 15–75 × 0.8–4.5 cm [13], up to 75 × 4 cm [14], 50–70(–90) × 2–6 cm [18] and up to 1 m × 6 cm [32]. Furthermore, from our study, we found the size of leaf blades ranges from 24–73.2 × 2–11.7 cm.,

644

sometimes wider than the previous studies.

sometimes wider than previous studies.

646–651

The previous studies reported the length of inflorescences of this species: 30–50 cm [10], 30–50 cm long or more [12], 30–60 cm [8,15,19], up to 60 cm [32], 30–70 cm [17] and 30–75 cm [11].

According to previous studies, the lengths of inflorescences are 30–50 cm [12], 30–50 cm long or more [14], 30–60 cm [10,17,20], up to 60 cm [32], 30–70 cm [18] and 30–75 cm [13]. In addition, we found the length of inflorescences ranges from 70–165 cm (including peduncle), 42–80 cm (excluding peduncle) in this study, sometimes longer than previous studies. The lengths of primary branches of inflorescences are 10–25 cm [20] and 20–30 cm (longest) [19], but we found (5–)10–43 cm long in this study,              sometimes longer than previous studies.

646–652

The lengths of primary branches of inflorescences of this species are 10–25 cm [19] and 20–30 cm (longest) [18], but we found (5–)10–43 cm long in this study, sometimes longer than the previous studies.

The lengths of primary branches of inflorescences are 10–25 cm [19] and 20–30 cm (longest) [18], but we found (5–)10–43 cm long in this study, sometimes longer than previous studies.

664–666

The previous studies reported the size of pollens of this species: 12 × 12 µm [33] and 29–40 µm in diam. [34].

According to previous studies, the sizes of pollens are 12 × 12 µm [33] and 29–40 µm in diam. [34]. Furthermore, from our study, we found the pollen diameter in equatorial axis ranges from 24.95–38.13 (29.52 ± 3.79) µm in consistent with Trigo & Fernández [34].

667–668

Sanghi & Sarna [33] reported the exine sculpturing of this species is psilate, but from our study, we found the granular.

According to Sanghi & Sarna [33], the exine sculpturing is psilate; however, from our observations, we found the granular.

671

A. donax

Arundo donax

672

Plant specimens of A. donax were observed and collected in the northern (Mae Hong Son, Chiang Mai, Lamphun and Nakhon Sawan) and the central……

Plant specimens of A. donax were observed and collected in the northern (Mae Hong Son, Chiang Mai, Lamphun and Nakhon Sawan), the eastern (Nakhon Ratchasima) and the central……

676–677

information added: The herbarium accession number can be seen in the specimens examined.

695

information added: The anatomy terminologies follow Metcalfe [39].

697

(Ngernsaengsaruay et al. Ad01-15122021)

(C. Ngernsaengsaruay et al. Ad01-15122021)

702–704

information added: The characteristics of pollen grains (size, shape, symmetry, aperture, and exine thickness and sculpturing) are examined and measured. The pollen morphology terminologies follow Punt et al. [40].

705–726

information added: see Conclusions

727

A. donax

Arundo donax

749

A. donax

Arundo donax

750–751

A. donax deposited in QBG.

A. donax deposited in QBG and Professor Dr Dokrak Marod for his own observation of A. donax in Kanchanaburi Province.

754

in the northern and the central regions

in the northern, the eastern and the central regions

Table A1.

p. 31

Ban Ai, Chai Prakan Subdistr., Chai Prakan Distr.,

Ban Ai, Si Dong Yen Subdistr., Chai Prakan Distr.,

Table A1.

p. 31

-

Information is added

References

The order of references from 6–20 was changed.

References

Reference no 39 and 40 are added.

Round 2

Reviewer 1 Report

The introduction was improved, but I still have doubts about the research question and objectives. This absence does not allow for following a logical line of the article's content. The discussion mentions that differences in plant growth and geographical distribution are affected by environmental factors, but which factors and how do they affect it?

Another important detail that worries me is that the discussion about anatomy is mentioned in the result, why? It does not make sense and is very confusing.

I am concerned that the conclusions do not correspond with the study performed, and the results are repeated. One conclusion is "A. donax is also cultivated as a medicinal grass, and is used in the treatment of itching rash" Where? How common is its use? What is the potential for commercialization?